# GauSAM: Contour-Guided 2D Gaussian Fields for Multi-Scale Medical Image Segmentation with Segment Anything

**Jinxuan Wu**[1]* **Jiange Wang**[1]* **Dongdong Zhang**[1]†

[1]School of Computer Science and Technology
Tongji University
{2154146, 2252699, ddzhang}@tongji.edu.cn

## Abstract

Effective multiscale medical image segmentation requires simultaneously preserving smooth spatial continuity and accurately delineating high-frequency boundaries, yet pixel-wise decoders often fail to maintain this balance consistently across varying resolutions. We introduce GauSAM, which seamlessly integrates contour-guided 2D Gaussian probability fields into the Segment Anything Model to address these challenges. In our framework, segmentation masks are parameterized as continuous probability fields of learnable 2D Gaussian primitives, enforcing spatially smooth and structurally consistent. Contourlet transforms extract rich multidirectional frequency information, notably edges and fine textures, which dynamically guide the spatial distribution of Gaussian primitives to substantially improve boundary fidelity in complex structures. The incorporation of these high-frequency contour priors also enriches the expressive capacity of the SAM image encoder. Extensive experiments on diverse 2D medical segmentation tasks confirm that GauSAM consistently delivers robust generalization and state-of-the-art performance with only 1.2M trainable parameters. The official implementation of GauSAM is publicly available at https://github.com/Quinten-Wu504/GauSAM.

## 1 Introduction

Medical image segmentation demands high precision and smooth boundary delineation, as anatomical structures often exhibit ambiguous edges and segmentation errors can lead to serious clinical consequences [1]. To address this, a wide range of methods have been proposed, including classical architectures such as U-Net [2] and several fine-tuned variants of the Segment Anything Model (SAM) [3]. These approaches typically rely on fixed-resolution, grid-based predictions and discrete interpolation schemes, which introduce discretization artifacts and limit their accuracy, especially in high-resolution or cross-scale scenarios [4, 5, 6].

To overcome the limitations of discrete segmentation, implicit neural representations (INRs) [7] have emerged as a resolution-agnostic alternative, mapping continuous spatial coordinates to feature representations. However, existing INRs still struggle to capture fine-grained structures due to their random sampling strategies, which often neglect local contours and directional textures. Furthermore, their reliance on discrete latent codes can obscure complex signals, ultimately reducing their ability to produce sharp and coherent segmentation outputs.

Motivated by the limitations of both discrete and INR-based methods, we propose to formulate segmentation as a continuous Gaussian probability field. We are the first to model SAM's segmentation

---

*These authors contributed equally to this work.
†Corresponding author.

39th Conference on Neural Information Processing Systems (NeurIPS 2025).

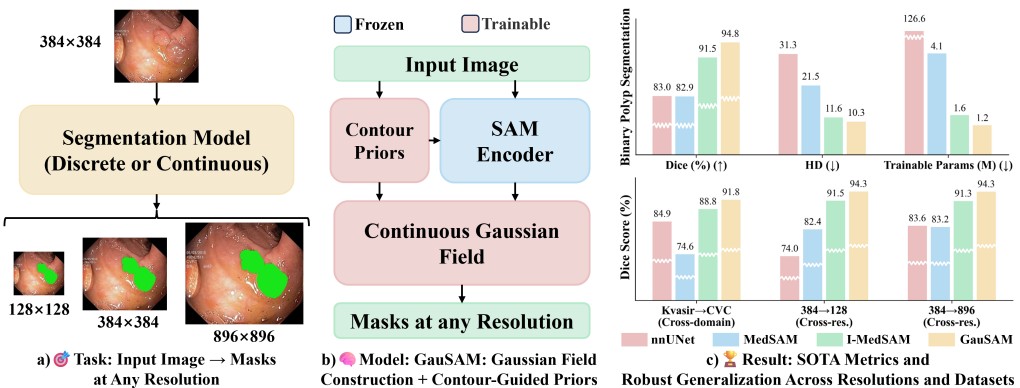

Figure 1: **Overview of GauSAM. a)** Illustration of resolution-agnostic mask generation: GauSAM processes a single image and produce segmentation masks at user-specified resolutions **b)** Schematic of GauSAM's architecture and workflow: an input image is encoded by a frozen SAM backbone, enhanced with contour priors derived from the Non-Subsampled Contourlet Transform (NSCT), and processed through a trainable continuous Gaussian field to generate resolution-agnostic segmentation masks. **c)** Summary of results: GauSAM achieves state-of-the-art Dice and Hausdorff Distance across in-domain, cross-domain, and cross-resolution settings, outperforming baselines.

probability field as a continuous representation parameterized by 2D Gaussian Splatting [8] for mask generation, enabling resolution-agnostic predictions while mitigating grid-related artifacts. Unlike prior works that employ Gaussian Splatting merely as auxiliary priors or post-processing components [9, 10, 11, 12], our framework is trained end-to-end, allowing the Gaussian parameters to be directly optimized with task supervision. However, standard Gaussian kernels lack directional sensitivity, often resulting in blurry or misaligned contours in the final mask result [13].

To address this issue, we propose Contour-Guided Gaussian-Field SAM (GauSAM) — a novel framework that enhances the SAM encoder with multiscale contour priors and constructs a contour-aware Gaussian field decoder. Specifically, GauSAM leverages the Non-Subsampled Contourlet Transform (NSCT) [14] to extract rich boundary features, which are progressively fused into the encoder via a Contour Propagation Unit (CPU). On the decoder side, we construct a resolution-agnostic probability field from a set of learnable Gaussian primitives, whose parameters are modulated by the contour-enhanced encoded features. A contour-guided spatial adjustment module further aligns these primitives to semantic boundaries, yielding high-fidelity masks with sharp contours and strong structural coherence.

Our main contributions are as follows:

1. We are the first to represent SAM's segmentation probability field as a continuous Gaussian probability field implemented via learnable 2D Gaussian Splatting, instantiated from a parameterized template library via Gumbel-Softmax sampling and modulated by contour-enhanced semantic features for stable training and compact representation.

2. We integrate NSCT-based structural priors into the SAM encoder via a lightweight Contour Propagation Unit, enhancing multiscale and multidirectional sensitivity to fine contours.

3. We propose a Contour-Guided Position Drift module to predict the spatial offsets of each Gaussian primitive from NSCT contour cues, improving boundary alignment.

4. GauSAM achieves superior performance over state-of-the-art discrete and INR-based methods across diverse 2D medical segmentation benchmarks, with notable robustness in cross-domain and resolution-agnostic inference.

## 2  Related Work

**2D Medical Image Segmentation**  Since the introduction of pixel-level segmentation by FCN [15], deep convolutional architectures such as U-Net [2] and its massive variant [16, 17, 18, 19]

have consistently achieved SOTA performance across a wide range of medical imaging modalities. More recently, the Vision Transformer–based Segment Anything Model (SAM) [3] has demonstrated remarkable zero-shot segmentation capabilities. To adapt SAM to the medical imaging domain, researchers have proposed various lightweight fine-tuning methods [20, 21, 22], all of which substantially improve the SAM encoder's ability to perceive medically relevant structures and features. However, these existing methods rely on fixed-resolution representations,which are prone to introducing discretization artifacts and blurred boundaries, especially in high-resolution or cross-scale scenarios [23]. To address these limitations, implicit neural representation (INR) methods [7, 24, 25] employ continuous neural fields to yield subpixel-smooth outputs at arbitrary resolutions, yet they still struggle with preserving precise texture structures and lacking interpretability [26].

**Contourlet-Based Enhancement**   The Contourlet transform achieves a multiscale, multidirectional subband decomposition via a cascade of a Laplacian pyramid and directional filter banks, efficiently capturing curves at arbitrary angles and high-frequency contour information [27]. Compared to traditional wavelets, Contourlets offer superior curve representation and directional selectivity, leading to widespread use in image and feature enhancement [27]. Leveraging these same properties, the non-subsampled Contourlet transform (NSCT) [14, 28, 29], which eschews downsampling via non-subsampled Laplacian pyramids and filter banks, achieves shift invariance and superior directional sensitivity compared to the conventional Contourlet transform. Recently proposed hybrid architectures that fuse Contourlet subband features with deep learning models [30, 31]have significantly enhanced the accuracy and robustness of fine-structure delineation, further validating the complementary role of frequency-domain enhancement in complex medical imaging tasks.

**Gaussian Splatting for Continuous Representation**   Gaussian Splatting (GS) explicitly represents a scene as a set of optimizable anisotropic Gaussian fields and leverages an efficient splatting-based rendering pipeline [8, 32]. In recent research, GS has shown broad potential across various vision tasks. In super-resolution reconstruction, GS-based approaches construct learnable 2D/3D Gaussian kernel radiance fields to enable real-time, high-fidelity interpolation at arbitrary upscaling factors [33, 34, 35, 36].And in semantic segmentation, emerging studies reveal that GS can reconstruct feature and radiance fields with superior precision — without per-pixel network inference — while preserving spatial and semantic consistency, thus offering a promising paradigm for efficient and accurate semantic segmentation [37, 38, 39]. Motivated by these advances, our work is the first to employ GS to construct continuous probability fields that map high-dimensional features to segmentation masks, preserving sharp boundaries and semantic consistency across scales.

## 3   Methodology

### 3.1   Preliminaries

Let $X \in \mathbb{R}^{H \times W \times C}$ denote the input image defined on its native discrete pixel lattice, where $H$, $W$, and $C$ denote the height, width, and number of channels respectively, and let $\Omega = [-1, 1]^2$ be the normalized continuous coordinate domain. We represent the segmentation target as a continuous mask probability field:

$$M : \ \Omega \ \rightarrow \ \Delta^{Class-1}, \quad \Delta^{Class-1} = \big\{ y \in \mathbb{R}^{Class} : \ y_c \geq 0 \ \forall c, \ \sum_{c=1}^{Class} y_c = 1 \big\} \tag{1}$$

where $Class$ is the number of semantic classes, and $\Delta^{Class-1}$ is the $(Class-1)$-simplex of class probabilities. The segmentation task is thus formulated as modeling a parametric mapping:

$$\mathcal{T}_\Theta : \ X \ \mapsto \ M. \tag{2}$$

By sampling the continuous field $M$ on an arbitrary target grid $p \subset \Omega$ of resolution $H' \times W'$, we obtain a discrete prediction $\widehat{O} \in \mathbb{R}^{H' \times W' \times C}$ that can be compared to ground-truth annotations using standard differentiable loss functions. Such a continuous representation of the probability field inherently supports arbitrary output resolutions and preserves spatial coherence.

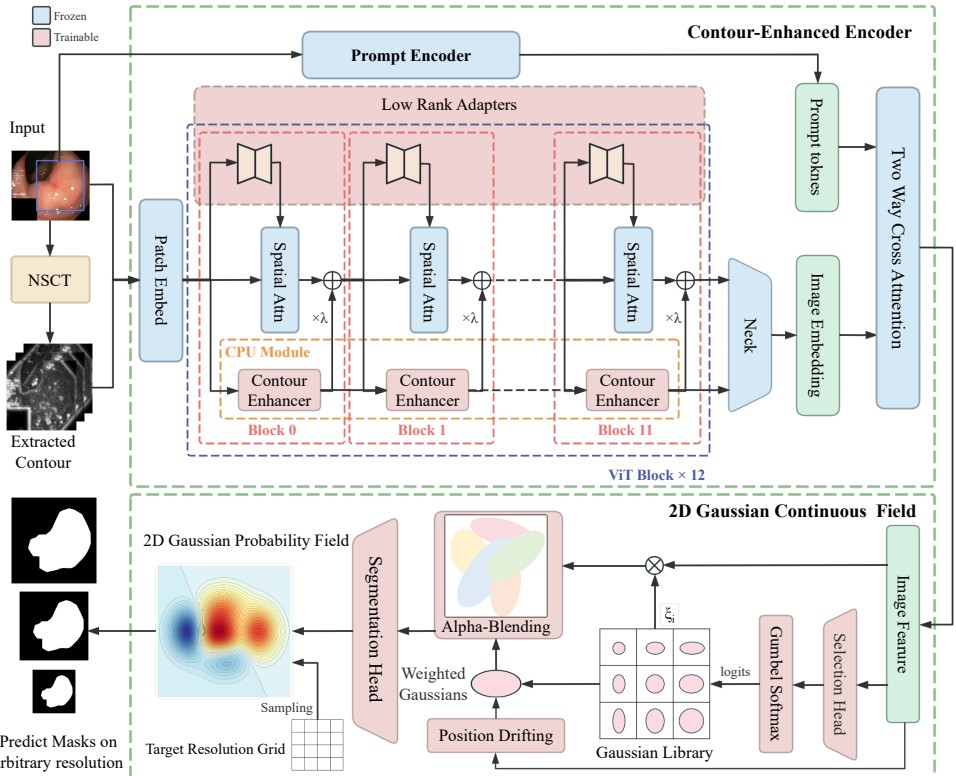

Figure 2: The overall architecture of GauSAM, consisting of Contour-Enhanced Encoding and 2D Gaussian Continuous Field Decoding. The input image is processed by a SAM image encoder, which is augmented by NSCT-based contour extraction and Contour Propagation Units (CPU). A continuous Gaussian field built from learnable 2D primitives then produces a resolution-agnostic segmentation mask. Red-colored modules indicate trainable components, including LoRA-adapted layers and newly proposed modules; blue modules denote frozen parts from the SAM-ViT-B backbone.

## 3.2 Overall Pipeline

As illustrated in Figure 2, GauSAM consists of two principal stages: (1) *Contour-Enhanced Image Encoding* and (2) *Continuous Field Decoding*.

In the first stage, the original SAM image encoder is enhanced with structural priors derived from a multi-level Non-Subsampled Contourlet Transform (NSCT) [14]. The resulting contour energy map $E$ is progressively injected into each transformer block via a lightweight Contour Propagation Unit (CPU), which recursively refines $E$ through adaptive modulation and soft fusion with semantic embeddings throughout the encoding hierarchy. In addition, the bounding-box prompt is encoded by SAM's prompt encoder as a sparse input, producing prompt embeddings that guide mask decoding through prompt-conditioned activations.

In the second stage, GauSAM constructs a continuous Gaussain probability field with weighted aggregation of $N$ learnable 2D anisotropic Gaussian primitives, each modulated by opacity-scaled encoder features. This approach forms a continuous representation $\mathcal{X}(p)$, which is decoded into a resolution-agnostic probability mask via a lightweight segmentation head. Moreover, each Gaussian primitive selects its covariance and opacity from a learned template library through a Gumbel-Softmax-based mechanism. A contour-Guided Position Drift module is also introduced to adjust the location of each primitive, using contour-aware offsets and enhancing boundary alignment without compromising semantic coherence.

We provide further details on the contour-guided encoder in Section 3.3, and on the Gaussian primitive field construction and decoding process in Section 3.4.

### 3.3 Contour-Enhanced Image Encoder

**Structural Contour Feature Extraction**   To enable high-fidelity modeling of fine-grained structures, we integrate into the image encoder a Structural Edge Extractor that leverages the Non-Subsampled Contourlet Transform (NSCT) to extract multiscale and multidirectional high-frequency contour features.

Given an input image $I \in \mathbb{R}^{B \times C \times H \times W}$, where $B$ denotes the batch size, we apply an NSCT decomposition with $L$ levels and a directional configuration $\Theta = \{\theta_1, \ldots, \theta_n\}$ at each level:

$$\{ContourSubband_{l,\theta}\}_{l=1,\theta=1}^{L,\Theta_l} = \mathcal{NSCT}(I) \tag{3}$$

To construct a unified structural prior without significantly increasing computational overhead, we aggregate all directional subband responses into a single contour-enhanced energy map $E$ for each channel. We apply a point-wise nonlinear transformation to each directional coefficient $ContourSubband_{l,\theta}$, combining logarithmic compression and gamma correction. The aggregated map is formulated as:

$$E = \sum_{l=1}^{L} \sum_{\theta=1}^{\Theta_l} \log \left(1 + |ContourSubband_{l,\theta}|\right)^{\gamma}, \quad \gamma \in (0,1) \tag{4}$$

The log space combination of coefficient maps, together with the gamma correction, reduces the dynamic range of contour features to suppress highlights, such as strong reflections caused by bodily fluids in rectal polyp images, thereby mitigating the impact of spurious strong contours on subsequent processing. The final structural map $E$ is subsequently normalized to emphasize weak contours, thus retaining essential structural signals across scales and orientations with minimal computational complexity.

**Contour Propagation Unit (CPU)**   To inject the extracted contour information into the ViT encoder in a progressive and semantically coherent manner, we design a novel Contour Propagation Unit (CPU). Instead of applying edge priors statically or at a single layer, CPU recursively propagates and refines contour information across the transformer hierarchy, ensuring that structural awareness is maintained through all encoding depths.

Each transformer block in the encoder receives two parallel inputs: the image embeddings $F_\ell$ and the contour feature embeddings $E_\ell$, which are derived from NSCT, where $\ell$ indexes the $\ell$-th transformer block from shallow to deep. Within each CPU-enhanced block, the contour signal is adaptively modulated through a lightweight MLP-based Edge Reweighting Module $\psi$:

$$R_\ell = \psi\big(\mathrm{LN}(E_\ell)\big), \quad E'_\ell = R_\ell \odot E_\ell \tag{5}$$

Here, $\mathrm{LN}(\cdot)$ denotes Layer Normalization, $R_\ell$ is a per-channel reweighting vector. The reweighted contour features are then softly fused into the image-embedding stream with a scaling parameter $\lambda$. Crucially, $E'_\ell$ is propagated to the next ViT block as $E_{\ell+1}$. This design introduces a structural inductive signal that complements the semantic representations in $F_\ell$, improving boundary fidelity across layers.

### 3.4 2D Gaussian Continuous Probability Field

To enable the construction of a continuous probability field, we construct a 2D continuous Gaussian feature field that inherently bridges the gap between discrete encoder outputs and continuous probabilistic representations. By stacking elliptical kernels in a continuous field, Gaussian Splatting enforces spatially smooth and locally consistent representations. This mathematically grounded mechanism aligns with the inherent spatial coherence of semantic segmentation, wherein adjacent pixels are likely to belong to the same semantic entity. Concretely, the feature field is formulated as the weighted superposition of N learnable Gaussian primitives. Each 2D Gaussian primitive is modeled as an anisotropic Gaussian distribution:

$$G_i\big(p \mid \mu_i, \Sigma_i\big) = \frac{1}{2\pi \, |\Sigma_i|} \exp\!\Big(-\frac{1}{2}\,(p - \mu_i)^\top \Sigma_i^{-1}(p - \mu_i)\Big), \quad p \in \Omega \tag{6}$$

where $i$ indexes the $i$-th Gaussian kernel, $\mu_i \in \Omega$ and $\Sigma_i \in \mathbb{R}^{2\times 2}$ denote the center and a positive-definite covariance of a primitive, respectively. In situations where multiple Gaussian primitives overlap, we modulate each discrete encoder feature $v_i \in \mathbb{R}^k$ by its learned opacity $\xi_i \in \mathbb{R}$ via $f_i = \sigma(\xi_i)\, v_i$, where $\sigma(\cdot)$ denotes the sigmoid function that maps $\xi_i$ into the interval $(0,1)$, serving as a soft opacity for each basis function. The final continuous feature field is then defined as the weighted aggregation of all Gaussian contributions:

$$\mathcal{X}(p) = \sum_{i=1}^{N} G_i(p \mid \mu_i, \Sigma_i) \cdot f_i \tag{7}$$

Subsequently, the blended field $\mathcal{X}(p)$ is forwarded through a lightweight segmentation head $g : \mathbb{R}^D \to \mathbb{R}^C$, producing class logits $g(\mathcal{X}(p))$. These logits are further normalized via a point-wise softmax operation to yield the continuous probability mask $M(p)$. By sampling $M(p)$ over arbitrary spatial grid, we obtain the final segmentation mask in a resolution-agnostic manner.

**Gaussian Parameters Library**  To address the optimization inefficiency in the above continuous feature field, associated with directly learning the covariance matrices $\Sigma_i$ and opacity scalars $\xi_i$ of the Gaussian primitives in an end-to-end approach, we introduce a learned library of $m$ template pairs: $\big\{(\Sigma^{(k)}, \xi^{(k)})\big\}_{k=1}^{m}$. Each primitive then derives its parameters by selecting from this library in a differentiable manner. Specifically, for the $i$-th Gaussian primitive located at position $p_i$ and associated with encoder feature $v_i$, a lightweight convolutional selection head projects $v_i$ into an $m$-dimensional logit vector:

$$z_i = \text{selHead}(v_i) \ \in \ \mathbb{R}^m. \tag{8}$$

To enable discrete selection while retaining differentiability, we apply the Gumbel-Softmax operator to $z_i$, producing a convex weight vector:

$$w_i = \text{GumbelSoftmax}(z_i) \ \in \ \Delta^{m-1} \tag{9}$$

which defines a probability distribution over the library items. The final parameters $\Sigma_i$ and $\xi_i$ for the $i$-th Gaussian primitive are then obtained as weighted combinations of the template items:

$$\Sigma_i = \sum_{k=1}^{m} w_{i,k}\, \Sigma^{(k)}, \quad \xi_i = \sum_{k=1}^{m} w_{i,k}\, \xi^{(k)} \tag{10}$$

This parameterization constrains scale, anisotropy, and opacity of each primitive to lie within a compact, pre-learned subspace of the full Gaussian parameter space, improving training stability while preserving the expressive richness required for detailed Gaussian-splatting representations [33].

**Contour-Guided Position Drift**  Existing Gaussian field approaches, owing to their uniformly distributed kernels, often struggle to capture boundary details adequately. We therefore propose a *Contour-Guided Position Drift* mechanism to enhance the contour representation in the Gaussian feature field. The core idea is to bias the placement of each Gaussian primitive toward semantic edges, thereby sharpening mask delineation without sacrificing spatial coherence. Specifically, for each initial pixel position $p_i$, we extract its contour feature vector $u_i \in \mathbb{R}^D$ from the *Contour-Guided Encoder* (Section 3.3), which contains adequate and multidirectional contour information. A lightweight MLP then predicts a small offset:

$$\Delta\mu_i = \tanh\big(\text{MLP}_{\text{off}}(u_i)\big) \tag{11}$$

and update the Gaussian center to $\mu_i = p_i + \Delta\mu_i$.

Inspired by prior work [34, 36] demonstrating the benefits for denser Gaussian placement in regions with richer textures, this drift mechanism ensures each primitive to align with structural boundaries identified in the contour feature space, while retaining consistency with the original semantic features.

### 3.5 Train GauSAM

We categorize the model parameters into two groups with distinct optimization strategies. The first group includes the pre-trained SAM image encoder and prompt encoder, whose parameters remain frozen to preserve their pretrained capacity. To adapt these pretrained components to our modified model, we introduce trainable LoRA adapters into the query $Q$ and value $V$ projection layers of each attention block, following prior work on efficient model adaptation of SAM [3]. The second group comprises all newly introduced modules, including the contour-guided feature extraction layers and the Gaussian-field decoder. These components are trained from scratch and fully optimized during training. The model is supervised using a pixel-wise *Dice–Cross-Entropy (DiceCE)* loss:

$$L_{\text{seg}}(o, \hat{o}) = \alpha\, L_{\text{CE}}(o, \hat{o}) + \beta\, L_{\text{Dice}}(o, \hat{o}), \tag{12}$$

where $L_{\text{CE}}$ measures the average cross-entropy across all $N$ pixels in the image, $L_{\text{Dice}}$ quantifies the spatial overlap between the predicted mask $\hat{o}$ and the ground-truth mask $o$, and $\alpha$ and $\beta$ balance their contributions.

## 4 Experiments

### 4.1 Datasets

We evaluate GauSAM on two representative benchmarks: Kvasir-Sessile [40] (196 sessile-polyp RGB images) for binary polyp segmentation and BCV [41] (30 CT slices with 13 organ classes) for multi-organ abdominal segmentation. To assess cross-domain generalization, models trained on Kvasir-Sessile are tested on CVC-ClinicDB [42] (612 RGB images from 31 endoscopic sequences), while those trained on BCV are evaluated on AMOS [43] (240 CT cases with liver segmentation only). All 3D CT volumes are processed as 2D axial slices. Data are split into training, validation, and testing sets with a 60:20:20 ratio. Input images are resized to $384 \times 384$ for polyp segmentation and $512 \times 512$ for multi-organ segmentation. To ensure fair comparison, we follow the preprocessing and evaluation protocols of I-MedSAM [25], reporting Dice Coefficient (Dice) and Hausdorff Distance (HD) on the test sets.

### 4.2 Implementation Details

**Gaussian Field Initialization**   To enable the continuous Gaussian field formulation, we initialize the Gaussian Parameter Library with 500 learnable templates. Each Gaussian primitive is parameterized by three elements of the covariance matrix $\Sigma$, including $\sigma_x, \sigma_y$ (linearly spaced between 0.2 and 3.0), and zero correlation, with an initial value of $\sigma(1) \approx 0.73$ for opacity $\xi$. Primitive centers are placed on a uniform grid, with position-drift offsets initialized to zero. Both the template parameters and the weights of the drift MLP employ Kaiming uniform initialization to ensure stable training.

**Model Architecture and Training**   We adopt the SAM-ViT-B backbone with LoRA adapters (rank 4) inserted into the query and value projections of each transformer block. The "up" projections use Kaiming uniform initialization, while the "down" projections are zero-initialized, ensuring early updates are confined to the LoRA modules, leaving the original SAM weights frozen. Only the LoRA-adapted layers and newly introduced modules are updated. A dual-branch learning rate strategy is used, with LoRA parameters trained at $0.05\times$ the rate of newly initialized ones. Optimization is performed using AdamW with a weight decay of $10^{-4}$ and a linear learning rate scheduler. The segmentation head is a lightweight MLP with two hidden layers of dimension 256. The Gaussian Parameters Library contains 500 learnable templates. Training is conducted on 4 NVIDIA H20 GPUs. The polyp segmentation task (single class) requires approximately 3 hours, while the multi-organ segmentation task takes around 50 hours. All models are trained for up to 1000 epochs with early stopping (patience = 10), using a hybrid DiceCE loss. Reported Dice and Hausdorff scores are evaluated on the test set using the checkpoint with the highest validation Dice.

Table 1: Segmentation results on binary polyp and multi-class organ tasks, compared with state-of-the-art discrete and Continuous approaches. Dice (%) denotes the mean Dice score ± standard deviation over 6 runs (fixed data splits, varying seeds). Params (M) indicates the number of trainable parameters, excluding frozen parameters that are not updated during optimization. Bold highlights the best performance in each task.

| Method | Binary Polyp Segmentation | | Multi-class Organ Segmentation | |
|---|---|---|---|---|
| | Dice (%)↑ | Params (M)↓ | Dice (%)↑ | Params (M)↓ |
| *Discrete Approaches* | | | | |
| U-Net [2] | 63.89±1.30 | 7.9 | 74.47±1.57 | 16.3 |
| Res2UNet [44] | 81.62±0.97 | 25.4 | 79.23±0.66 | 38.3 |
| nnUNet [16] | 82.97±0.89 | 126.6 | 85.15±0.67 | 126.6 |
| MedSAM [45] | 82.88±0.55 | 4.1 | 85.85±0.81 | 52.7 |
| *Continuous Approaches* | | | | |
| OSSNet [46] | 76.11±1.14 | 5.2 | 73.38±1.65 | 7.6 |
| IOSNet [7] | 78.37±0.76 | 4.1 | 76.75±1.37 | 6.2 |
| SwIPE [47] | 85.05±0.82 | 2.7 | 81.21±0.94 | 4.4 |
| I-MedSAM [25] | 91.49±0.52 | 1.6 | **89.91±0.68** | 3.5 |
| **GauSAM(Ours)** | **94.76±0.32** | **1.2** | 89.75±0.47 | **2.9** |

## 4.3 Quantitative Results

**Comparison with State-of-the-Art Methods**   We evaluate GauSAM on two representative benchmarks: the Kvasir-Sessile dataset for binary polyp segmentation, and the BCV dataset for multi-class abdominal organ segmentation. The compared methods are grouped into two categories: discrete approaches, which predict segmentation masks on fixed grids, and continuous approaches, which represent masks in resolution-agnostic or implicit forms. Discrete baselines include U-Net [2], Res2UNet [44], nnUNet [16], and MedSAM [45]. Continuous baselines include OSSNet [46], IOS-Net [7], SwIPE [47], and I-MedSAM [25] —— the latter replaces the original decoder of SAM with an INR–based formulation, and represents the previous state-of-the-art.

As demonstrated in Table 1, GauSAM consistently achieves the best performance across both tasks, obtaining the highest Dice scores while maintaining the lowest number of trainable parameters. On the Kvasir-Sessile dataset, GauSAM reaches a Dice of **94.76%**, outperforming I-MedSAM [25] by over 3 points with fewer parameters (1.2M vs. 1.6M). On the multi-organ task, GauSAM achieves a Dice of **89.75%**, which is on par with I-MedSAM, despite using only 2.9M parameters.

**Cross-Domain Generalization Performance**   We evaluate the cross-domain generalizability of GauSAM under two settings: binary-class polyp segmentation from Kvasir-Sessile to CVC, and from multi-organ segmentation in BCV to liver-only segmentation in AMOS. All models are trained on the source domain and directly evaluated on the target domain without any fine-tuning. As shown in Table 2, GauSAM achieves the highest Dice scores across both transfer scenarios, reaching **91.79%** on CVC and **90.62%** on AMOS. Compared with discrete and continuous baselines, GauSAM exhibits superior generalization under domain shifts. In particular, it consistently outperforms the previous state-of-the-art I-MedSAM [25] and other baselines.

**Cross-Resolution Generalization Performance**   To evaluate the resolution-agnostic capability of GauSAM, we conduct cross-resolution segmentation experiments on the Kvasir-Sessile dataset. All models are trained with input images and masks at $384 \times 384$, and are then evaluated by producing output masks at $128 \times 128$ and $896 \times 896$, without any fine-tuning or architectural modification. As shown in Table 3, GauSAM consistently achieves best performance under both downsampling and upsampling conditions, reaching Dice scores of **94.35%** and **94.29%**, respectively. In contrast, discrete baselines exhibit notable performance drops when deployed at unseen resolutions, and even implicit models such as SwIPE [47] and I-MedSAM [25] show limited adaptability.

**Boundary Comparison**   Hausdorff Distance (HD) measures boundary accuracy by capturing the largest surface deviation between the prediction and the ground truth. As shown in Table 4, GauSAM achieves strong performance across most settings, particularly in cross-resolution scenarios. A relatively weaker HD score is observed in the domain adaptation from Kvasir to CVC; however, a significant improvement is still evident in the corresponding Dice score as shown in Table 2. In the

Table 2: Cross-domain segmentation results from Kvasir-Sessile to CVC and from BCV to AMOS.

| Method | Dice (%)↑ |
|---|---|
| *Kvasir-Sessile → CVC* | |
| nnUNet [16] | 84.91 |
| MedSAM [45] | 74.59 |
| I-MedSAM [25] | 88.83 |
| **GauSAM (Ours)** | **91.79** |
| *BCV → AMOS* | |
| nnUNet [16] | 79.63 |
| MedSAM [45] | 71.98 |
| I-MedSAM [25] | 86.28 |
| **GauSAM (Ours)** | **90.62** |

Table 3: Cross-resolution segmentation results on Kvasir-Sessile from 384 to 128 and 384 to 896. (*: adjusted input resolution)

| Method | Dice (%)↑ | |
|---|---|---|
| | $384 \to 128$ | $384 \to 896$ |
| *Discrete Approaches* | | |
| nnUNet [16] | 73.97 | 83.56 |
| nnUNet* [16] | 65.34 | 76.36 |
| MedSAM [45] | 82.39 | 83.19 |
| MedSAM* [45] | 82.37 | 83.32 |
| *Implicit Approaches* | | |
| SwIPE [47] | 81.26 | 84.33 |
| I-MedSAM [25] | 91.45 | 91.33 |
| **GauSAM (Ours)** | **94.35** | **94.29** |

Table 4: Hausdorff Distance comparison on various experiment settings.

| HD distance (↓) | Kvasir-Sessile | Kvasir-Sessile → CVC | $384 \to 128$ | $384 \to 896$ | BCV | BCV → AMOS |
|---|---|---|---|---|---|---|
| nnUNet [16] | 31.30 | 82.31 | 13.69 | 72.31 | 6.50 | 80.39 |
| MedSAM [45] | 21.53 | 30.15 | 8.04 | 51.82 | 10.62 | 52.14 |
| I-MedSAM [25] | 11.59 | **19.76** | 7.91 | 32.77 | **5.95** | 37.53 |
| **GauSAM (Ours)** | **10.29** | 20.43 | **4.21** | **22.94** | 11.21 | **36.51** |

Table 5: End-to-end inference performance comparison. Parameters (Params) denote the total number of model parameters in millions (M). GFLOPs represent giga floating-point operations. Latency is the average inference time in milliseconds (ms) with standard deviation (STD). FPS indicates frames per second. Memory refers to peak GPU memory usage in megabytes (MB). All metrics were measured on a single NVIDIA RTX 4090 GPU with batch size 1 and input resolution $512 \times 512$, averaged over 100 forward passes after 10 warm-ups.

| Model | Params (M) | GFLOPs | Latency (ms) $\pm$ STD | FPS | Memory (MB) |
|---|---|---|---|---|---|
| U-Net | **31.03** | **218.97** | **41 $\pm$ 0.06** | **24.4** | **1244** |
| MedSAM | 93.74 | 573.87 | 66 $\pm$ 0.91 | 15.2 | 2768 |
| I-MedSAM | 92.88 | 648.06 | 86 $\pm$ 3.12 | 11.6 | 3021 |
| **GauSAM (Ours)** | 92.51 | 473.35 | 188 $\pm$ 2.34 | 5.3 | 1898 |
| GauSAM w/o NSCT | 92.51 | 374.91 | 79 $\pm$ 2.15 | 12.7 | 1898 |

BCV source-domain setting, GauSAM does not perform well, which may be attributed to the use of a single continuous Gaussian field for all organ classes, potentially limiting precision on small structures in multi-organ segmentation.

**Computational Efficiency** To evaluate the computational efficiency of GauSAM, we benchmark it against several representative baselines. As demonstrated in Table 5, GauSAM exhibits lower computational efficiency, while delivering superior segmentation performance. We emphasize that this inference speed is well within the practical range for medical applications, fully meeting the real-time demands of clinical workflows. In medical image segmentation tasks, precision is always paramount, as accurate delineation of anatomical structures or pathological regions directly impacts diagnostic accuracy and patient safety. Moreover, empirical evidence shows that removing the NSCT module leads to a substantial drop in GFLOPs and latency, confirming that this component is the main contributor to GauSAM's computational overhead. The Appendix D provides component-wise efficiency metrics and a diagnostic analysis of the NSCT module's comparatively lower efficiency.

## 4.4 Qualitative Results

Figure 3 presents representative qualitative comparisons on both binary and multi-class segmentation tasks. We visualize the predicted segmentation masks of different models by overlaying them on the

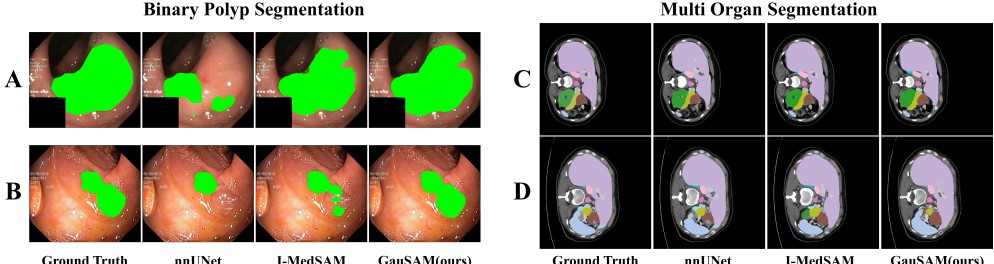

Figure 3: Qualitative comparison on binary polyp segmentation and multi-organ segmentation.

original input images. Compared with existing discrete and implicit methods, GauSAM produces masks that exhibit significantly improved semantic smoothness and structural consistency. These improvements are particularly evident in $SampleA$, where GauSAM demonstrates superior edge continuity along the polyp's right boundary compared to the previous SOTA method, I-MedSAM, which produces fragmented artifacts. In contrast, GauSAM maintains consistent boundaries without such discontinuities. These results are largely attributed to our continuous Gaussian field formulation, which leverages elliptical Gaussian kernel to ensure spatial continuity and local consistency, meaning that adjacent pixels are more likely to belong to the same object. This principled design, combined with contour-guided structural cues, enables the model to better capture fine-grained geometry and align predictions with anatomical boundaries, effectively mitigating discretization artifacts and enhancing the coherence of structural delineation across diverse regions.

### 4.5 Ablation Study

Our core contribution lies in the design of the continuous Gaussian Field (GS), upon which all subsequent modules are built. Therefore, we conduct ablation studies on the Kvasir-Sessile dataset under three settings: in-domain, cross-domain, and cross-resolution. As shown in Table 6, the baseline model without GS performs significantly worse across all tasks. After introducing the Gaussian Field alone, the model yields substantial improvements, validating its critical role. Further integrating the Contour Propagation Unit (CPU) enhances spatial consistency, while the Contour-Guided Position Drift (PosOff) improves robustness under resolution shifts. The full model, which includes all three components, consistently achieves the highest Dice scores in every setting.

Table 6: Ablation study on the effectiveness of each component in the pipeline. Dice(%) is reported on in-domain, cross-domain, and cross-resolution settings.

| GS | CPU | PosOff | Kvasir-Sessile | Cross-domain Kvasir → CVC | Cross-resolution 384 → 128 | 384 → 896 |
|----|-----|--------|----------------|---------------------------|----------------------------|-----------|
| – | – | – | 86.77 | 84.51 | 75.02 | 77.43 |
| ✓ | – | – | 92.62 | 91.00 | 91.44 | 92.74 |
| ✓ | ✓ | – | 93.62 | 91.07 | 92.33 | 94.00 |
| ✓ | ✓ | ✓ | **94.76** | **91.79** | **94.35** | **94.29** |

## 5 Conclusion and Discussion

In this paper, we presents GauSAM, a novel segmentation framework that combines contour-enhanced encoding with continuous Gaussian field decoding. By constructing a continuous resolution-agnostic Gaussian probability field, our method addresses limitations of structural precision and boundary localization of both discrete and INR-based approaches. Extensive experiments across binary and multi-class segmentation, under in-domain, cross-domain, and cross-resolution settings, demonstrate GauSAM's superior performance in terms of semantic consistency and edge accuracy. However, GauSAM underperforms on multi-class segmentation benchmarks, likely due to its use of a single continuous field to represent multiple classes, which may obscure fine-grained details of small objects. Despite this, GauSAM is the first to model SAM's segmentation probability field as a continuous representation parameterized by 2D Gaussian Splatting for mask generation and establishes a strong foundation for future improvements in multi-class scenarios.

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

# A    Additional Analysis of NSCT and Related Modules

The integration of the Non-Subsampled Contourlet Transform (NSCT) into the GauSAM framework plays a pivotal role in enhancing the segmentation accuracy of the Segment Anything Model (SAM) for both multi-organ and binary polyp segmentation tasks. The choice of NSCT and the design of related modules stems from a meticulous combination of theoretical analysis and extensive experimental validation, as detailed in the following subsections.

## A.1    Formal Analysis of Contour-Guided Position Drift

The Contour-Guided Position Drift module improves boundary alignment by applying bounded, differentiable shifts to the centers of Gaussian kernels. For the $i$-th anisotropic Gaussian kernel:

$$G_i(p \mid \mu_i, \Sigma_i) = \frac{1}{2\pi|\Sigma_i|} \exp\left(-\frac{1}{2}(p - \mu_i)^\top \Sigma_i^{-1}(p - \mu_i)\right), \tag{13}$$

its center is updated from the initial position $p_i$ by:

$$\mu_i = p_i + \Delta\mu_i, \quad \Delta\mu_i = \tanh\left(\mathrm{MLP}_{\mathrm{drift}}(u_i)\right), \tag{14}$$

where $\tanh(\cdot)$ ensures $\|\Delta\mu_i\| < 1$ for stability. During training, according to the chain rule, the gradient of the segmentation loss $L$ with respect to $\mu_i$ is:

$$\frac{\partial L}{\partial \mu_i} = \int_\Omega \frac{\partial L}{\partial X(p)} \frac{\partial X(p)}{\partial G_i} \frac{\partial G_i}{\partial \mu_i} \, \mathrm{d}p, \tag{15}$$

with

$$\frac{\partial G_i(p)}{\partial \mu_i} = \Sigma_i^{-1}(p - \mu_i) \, G_i(p). \tag{16}$$

This formulation indicates that a small adjustment of $\mu_i$ redistributes the kernel's density in a distance-weighted manner, directly influencing how features are aligned to local structures. Through the fully differentiable mapping $\Delta\mu_i = \tanh(\mathrm{MLP}_{\mathrm{drift}}(u_i))$, these gradients propagate to the parameters of the drift network, enabling the drift module to automatically optimize kernel placement. Consequently, the drift module is a mathematically grounded, bounded, and differentiable mechanism that, under gradient descent, provably minimizes boundary misalignment, complementing NSCT's precise contour extraction.

## A.2    Qualitative Analysis of NSCT-Processed Edge Maps

To further elucidate the contribution of NSCT, we present an additional qualitative analysis based on the edge maps generated from NSCT processing, as illustrated in Figure 4. The provided image showcases a comparative visualization of original images and their corresponding NSCT-processed edge maps across two distinct medical imaging modalities: CT scans for multi organ segmentation (rows 1-2) and endoscopic images for binary polyp segmentation (rows 3-4). In the original CT images (row 1), organ boundaries exhibit significant ambiguity due to low contrast and overlapping structures, posing challenges for precise delineation. Similarly, the original endoscopic images (row 3) reveal blurred polyp edges, often confounded by background noise and varying illumination. The application of NSCT, however, effectively transforms these inputs into well-defined edge maps (rows 2 and 4), where anatomical boundaries——such as those between organs or around polyps——are sharply delineated with enhanced multidirectional frequency information. This enhancement lays a solid foundation for GauSAM's encoding process in terms of edge information representation and precise Gaussian kernel distribution, a benefit also reflected in the ablation study results, particularly in cross-domain and cross-resolution tasks.

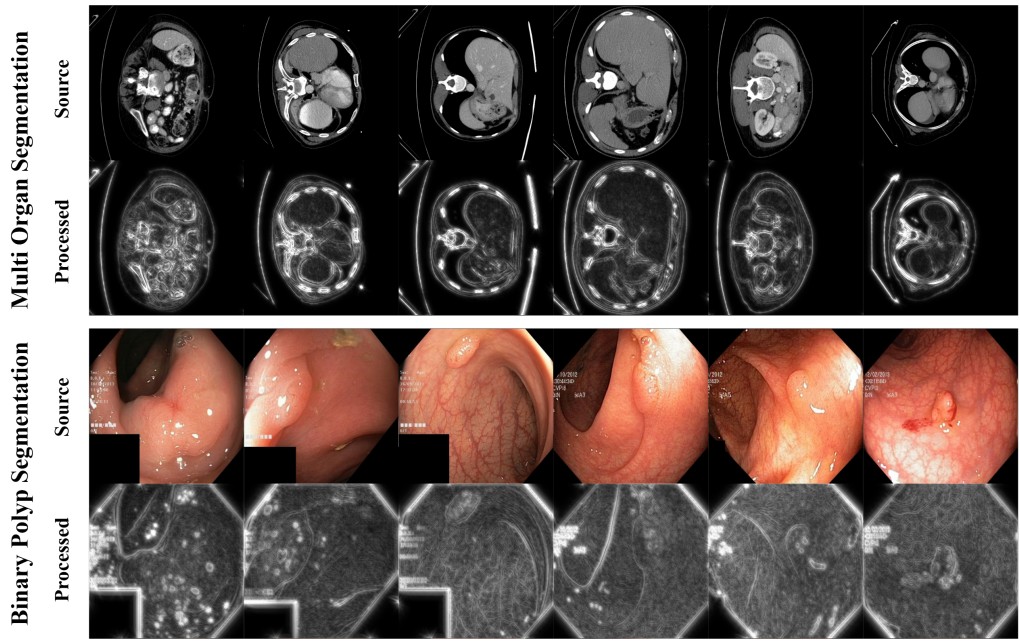

Figure 4: Qualitative comparison of source images (top) and their corresponding NSCT-processed edge maps (bottom) for multi-organ segmentation (rows 1-2) and binary polyp segmentation (rows 3-4). The NSCT-processed edge maps enhance boundary clarity, improving SAM's encoder edge representation and guiding the decoder's Gaussian kernel distribution.

Table 7: Symbol definitions for Gaussian Parameters Library and traditional Gaussian Splatting.

| Symbol | Definition |
|--------|------------|
| $i$ | Index of each Gaussian kernel |
| $k$ | Index of each template in the library |
| $N$ | Number of Gaussian kernels |
| $m$ | Number of templates in the library ($N \gg m$) |
| $\Sigma_i$ | $2 \times 2$ symmetric covariance matrix for kernel $i$, controlling ellipsoid shape and orientation (3 parameters) |
| $\xi_i$ | Opacity of kernel $i$ (1 parameter) |
| $\Sigma^{(k)}$ | Covariance matrix of template $k$ |
| $\xi^{(k)}$ | Opacity of template $k$ |
| $\theta_i$ | Parameters of kernel $i$, including $\Sigma_i$ and $\xi_i$ |
| $\theta$ | All parameters of traditional method, $\{\theta_i\}_{i=1}^{N}$ |
| $\phi$ | Library parameters, $\{(\sigma_x/\sigma_y, \mathrm{corr}, \xi^{(k)})\}_{k=1}^{m}$ |

## B  Detailed Analysis of Gaussian Parameters Library

To provide a comprehensive understanding of the Gaussian Parameters Library, we present a formal comparison with the traditional Gaussian Splatting method, highlighting the theoretical and practical advantages of our approach. The adoption of the Gaussian Parameters Library is grounded in an extensive literature review, rigorous theoretical derivations, and comprehensive experimental validation, as evidenced by the ablation study on library size reported in Section E.2. The library approach achieves superior training efficiency and stability compared to traditional Gaussian Splatting, which we formalize below. Due to fundamental architectural differences, direct ablation of the library is impractical, as it would necessitate a complete restructuring of the model. The following subsections define the key symbols and provide a detailed comparison of the two approaches.

## B.1 Symbol Definitions

Table 7 defines the symbols used in the analysis of the Gaussian Parameters Library and traditional Gaussian Splatting.

## B.2 Comparison with Traditional Gaussian Splatting

The Gaussian Parameters Library optimizes the covariance matrix $\Sigma_i$ and opacity $\xi_i$ of Gaussian kernels with greater stability and efficiency than traditional Gaussian Splatting. Below, we compare the two approaches across several key aspects, focusing on parameter count, parameter space, initialization and training stability, learning objective, and model structure.

**Parameter Count**  Classical Gaussian Splatting constructs a continuous field by overlapping many elliptical kernels so that, after opacity accumulation, the synthesized values at discrete pixels/feature locations match the observed signal. Training therefore adjusts the kernels' shapes and opacities to a specific image so as to best reconstruct its discrete measurements, i.e., an instance-specific optimization without parameter sharing across images. In practice, the number of kernels $N$ scales with the number of feature-space locations (often one kernel per pixel/voxel/patch), so the parameter size grows with image resolution/content. Without any library/shared parameterization, this instance-specific paradigm necessitates retraining for each new image, which is computationally prohibitive at scale.

Formally, if each of the $N$ kernels uses a $2 \times 2$ symmetric covariance $\Sigma_i$ (3 parameters) and an opacity $\xi_i$ (1 parameter), the parameter count is

$$P_{\text{trad}} = (3+1) \cdot N = 4N. \tag{17}$$

By contrast, we introduce a template library with $m$ shared kernels and a lightweight selection head, which inherently form a hypernetwork. Given image embeddings, the hypernetwork maps image features directly to each kernel's parameters by selecting (and modulating) a template for that location, thus enabling runtime parameter prediction and avoiding instance-specific retraining. The library holds only $m$ sets of covariance/opacity, yielding

$$P_{\text{lib}} = (3+1) \cdot m = 4m. \tag{18}$$

Since typically $N \gg m$, we have $P_{\text{trad}} \gg P_{\text{lib}}$, achieving a substantial reduction in parameter size while enabling end-to-end learning with shared parameters across images.

**Parameter Space**  The traditional method optimizes parameters in a high-dimensional space

$$S_{\text{trad}} = \mathbb{R}^{4N}, \tag{19}$$

whereas the library method operates in

$$S_{\text{lib}} = \mathbb{R}^{4m}. \tag{20}$$

Given $N \gg m$, the lower-dimensional space of the library method facilitates faster convergence and reduces computational complexity.

**Initialization and Training Stability**  The traditional method employs random initialization for $\theta_i = (\Sigma_i, \xi_i)$, which risks degenerate covariance matrices (i.e., $\det(\Sigma_i) \approx 0$), causing ellipse collapse and unstable gradients. The library method initializes

$$\phi = \{(\sigma_x/\sigma_y) \in [0.2, 3.0], \text{corr} = 0, \xi^{(k)} = \text{sigmoid}(1)\} \tag{21}$$

with Kaiming uniform initialization and uses Gumbel-Softmax to select from $m$ positive definite templates $\{\Sigma^{(k)}\}_{k=1}^{m}$. This ensures valid $\Sigma_i$ values, enhancing initialization stability and training robustness.

**Learning Objective and Training Efficiency**  The traditional method optimizes the loss

$$L_{\text{trad}} = \min_{\theta \in \mathbb{R}^{4N}} L(\theta), \tag{22}$$

which is computationally intensive due to the high-dimensional space. The library method optimizes

$$L_{\text{lib}} = \min_{\phi \in \mathbb{R}^{4m}} L(\phi), \tag{23}$$

where a selection head with Gumbel-Softmax assigns one of the $m$ templates to each kernel, operating in the lower-dimensional space $\mathbb{R}^{4m}$. This constrained optimization accelerates convergence and improves training efficiency.

## C   Evaluation Results on Additional Datasets

Due to strict page constraints in the main manuscript, we presented quantitative results only on the Kvasir-Sessile and BCV datasets, selected for their representativeness and recognized challenges in medical image segmentation. To further demonstrate the robustness of GauSAM, we conducted extensive experiments across additional medical imaging modalities and segmentation tasks. In this section, we report supplementary results on the REFUGE2 dataset (color fundus images for optic cup and disc segmentation) and the Breast Ultrasound Image (BUSI) dataset (lesion segmentation), which further validate GauSAM's state-of-the-art performance across diverse tasks and modalities. As shown in Table 8, GauSAM consistently outperforms baseline methods, achieving the highest Dice scores across all tasks. On the REFUGE2 dataset, GauSAM attains Dice scores of $89.1\%$ for optic cup segmentation and $96.2\%$ for optic disc segmentation, surpassing the previous state-of-the-art method, I-MedSAM, by 1.2% and 2.2%, respectively. Similarly, on the BUSI dataset, GauSAM achieves a Dice score of 92.16% for lesion segmentation, outperforming I-MedSAM by 0.61%. These results confirm GauSAM's superior robustness in handling diverse medical imaging challenges.

Table 8: Performance comparison on REFUGE2 and BUSI datasets. Dice (%) is reported for each task, evaluated on the respective test sets.

| Model | Optic Cup Dice (%) | Optic Disc Dice (%) | BUSI Lesion Dice (%) |
|---|---|---|---|
| U-Net [2] | 82.3 | 92.1 | 67.24 |
| nnUNet [16] | 84.9 | 94.7 | 75.19 |
| I-MedSAM [25] | 87.9 | 94.0 | 91.55 |
| **GauSAM (Ours)** | **89.1** | **96.2** | **92.16** |

## D   Detailed Computational Efficiency Analysis of GauSAM Components

To further elucidate the computational efficiency of GauSAM, we provide a detailed decomposition of its component-level inference performance in Table 9. The table presents the computational metrics for each major component of GauSAM, and the evaluation protocol is identical to that of Section 4.3. As shown in Table 9, the NSCT module accounts for approximately 58% of GauSAM's total latency (109 ms out of 188 ms), making it the primary contributor to the model's computational overhead.

This phenomenon can be attributed to two key factors:

1. **NSCT Feature Richness vs. Computational Cost:** The NSCT module excels in capturing multi-scale and multi-directional contour features, which significantly enhance segmentation accuracy (e.g., a 1.08% Dice improvement over FFT, 94.76% vs. 93.68%). However, its shift-invariant filter bank decomposition across multiple scales and orientations inherently increases computational complexity, leading to higher GFLOPs and latency.

2. **Implementation Limitations:** Existing NSCT implementations are outdated and limited to Python 2.x, lacking compatibility with modern architectures. Our custom Python 3.x implementation, while supporting vectorized processing for batch sizes greater than 1, lacks

the low-level optimizations (e.g., C++ or CUDA-based libraries) available in more mature algorithms, resulting in suboptimal execution efficiency.

These findings confirm that the computational overhead in GauSAM is primarily due to the NSCT module's intrinsic complexity and suboptimal implementation. Nevertheless, the enhanced contour accuracy justifies this trade-off, as it ensures robust representational quality critical for medical imaging while maintaining clinically acceptable throughput.

Table 9: Component-wise inference profile of GauSAM, under the same metric definitions and evaluation protocol as Table 5.

| Component | Params (M) | GFLOPs | Latency (ms) $\pm$ STD | FPS | Memory (MB) |
|---|---|---|---|---|---|
| NSCT | $\approx 0$ | 98.44 | $109.01 \pm 1.82$ | 9.2 | 113 |
| Contour-Enhanced Encoder | 92.13 | 220.18 | $43.53 \pm 0.79$ | 23.0 | 388 |
| 2D Gaussian Continuous Field | 0.38 | 154.73 | $32.70 \pm 1.15$ | 30.6 | 1397 |
| **GauSAM (end-to-end)** | 92.51 | 473.35 | $188.01 \pm 2.34$ | 5.3 | 1898 |

# E  Additional Ablation Studies

## E.1  Contour Extraction Ablation

To validate GauSAM's choice of the Non-Subsampled Contourlet Transform (NSCT) for contour feature extraction, we conduct ablation studies on the Kvasir-Sessile dataset, replacing NSCT with alternative contour/high-frequency methods: FFT (as used in I-MedSAM [25]), Canny, Wavelet, and Sobel. As shown in Table 10, the full GauSAM with NSCT achieves the highest Dice score and lowest Hausdorff Distance (HD), confirming its superiority in precise boundary delineation. Unlike FFT's broad high-frequency components that include noise, NSCT selectively captures semantically meaningful contours, aligning directly with the requirements of semantic segmentation and providing superior boundary representation. These experiments confirm NSCT's superiority in precise boundary delineation and its better compatibility with GauSAM.

Table 10: Ablation study on contour feature extraction methods on Kvasir-Sessile.

| Method | Dice (%) | HD |
|---|---|---|
| FFT | 93.68 | 11.34 |
| Canny | 93.66 | 11.82 |
| Wavelet | 94.13 | 11.40 |
| Sobel | 93.71 | 11.15 |
| **NSCT (Ours)** | **94.76** | **10.29** |

## E.2  Impact of Gaussian Parameter Library Size

As reported in Table 11, expanding the Gaussian Parameter library from $m = 50$ to $m = 500$ leads to a marked improvement in segmentation accuracy and boundary precision.This improvement indicates that a richer set of 2D Gaussian templates better captures the diverse scale and shape variations in our medical image benchmarks. Beyond $m = 500$, however, the benefit plateaus and slightly reverses, indicating diminishing returns once the library exceeds a critical size.

A primary factor behind this saturation is the linear growth of the selection head's parameters with $m$. Since the module must assign a probability to each template via a soft selection mechanism, overly large $m$ introduces many logits that receive relatively weak gradient signals, leading to under-utilized or poorly distinguished templates and impeding stable convergence during training.

In addition, an excessively large library incurs higher computational and memory costs at both training and inference time, while introducing substantial redundancy: many Gaussian templates end up modeling similar feature patterns, so their incremental contribution becomes marginal. Based on a series of controlled experiments, we find $m = 500$ strikes the best balance between expressive power and learnability, yielding the highest average Dice score with moderate overhead.

Table 11: Segmentation performance under different Gaussian parameter library sizes on the Kvasir-Sessile dataset.

| Library size $m$ | 50 | 100 | 300 | 500 (default) | 800 | 1000 |
|---|---|---|---|---|---|---|
| Dice (%) | 92.93 | 93.78 | 94.21 | **94.76** | 94.22 | 94.43 |
| HD Distance | 12.53 | 11.65 | 11.01 | **10.29** | 11.19 | 10.93 |
| Params (M) | **0.96** | 0.99 | 1.11 | 1.22 | 1.40 | 1.51 |

# F   Supplementary Analysis of Robustness on Low-Quality Images

To further validate its robustness, particularly on low-quality images with blurred boundaries, we conducted additional experiments on the Kvasir-Sessile dataset, comparing GauSAM against the previous state-of-the-art model, I-MedSAM [25]. This analysis provides comprehensive empirical evidence of GauSAM's superior performance across varying image quality levels. We identified 12 test images where I-MedSAM exhibited relatively low performance ($Dice < 89\%$), all of which displayed ambiguous boundaries. Based on this observation, we partitioned the test set into two groups: a *High Ambiguity* subset, comprising images with blurred or uncertain boundaries, and a *Normal* subset, consisting of images with clear, well-defined boundaries. Table 12 presents the mean Dice scores for both models on these subsets.

As shown in Table 12, GauSAM achieves a substantial Dice improvement of $+7.3\%$ over I-MedSAM on the High Ambiguity subset, compared to a $+1.9\%$ improvement on the Normal subset. This pronounced performance gain in the ambiguous boundary scenario underscores GauSAM's robustness in handling low-quality images with blurred boundaries. Furthermore, GauSAM exhibits remarkable consistency across image quality levels, with only a $0.5\%$ Dice score difference between the High Ambiguity and Normal subsets, in contrast to I-MedSAM's $6\%$ gap. These results confirm GauSAM's ability to maintain stable and accurate segmentation performance across diverse image quality conditions, reinforcing its effectiveness for both challenging and well-defined cases.

Table 12: Dice performance comparison of GauSAM and I-MedSAM on the Kvasir-Sessile test set, divided into High Ambiguity and Normal image groups. Evaluations were conducted with a batch size of 1 and input resolution of $512 \times 512$. Dice Gain represents the improvement of GauSAM over I-MedSAM.

| Image Type | Sample Count | I-MedSAM Dice (%) | GauSAM Dice (%) | Dice Gain (%) |
|---|---|---|---|---|
| High Ambiguity | 11 | 86.8 | **94.1** | +7.3 |
| Normal | 28 | 92.7 | **94.6** | +1.9 |

# G   Extension to 3D Segmentation Tasks

The design of GauSAM, particularly its use of 2D Gaussian Splatting and contour-guided mechanisms, is inherently extensible to 3D volumetric segmentation. This section provides a formal analysis of how GauSAM's framework can be adapted to 3D tasks, highlighting its potential to enable expressive and spatially coherent volumetric representations while preserving fine-grained anatomical details. The following subsections extend the core components of GauSAM——specifically the Non-Subsampled Contourlet Transform (NSCT) and the Gaussian Continuous Probability Field——to their 3D counterparts, demonstrating the theoretical compatibility and practical promise of this extension.

## G.1   3D Non-Subsampled Contourlet Transform

In 3D, the input volume is represented as $I \in \mathbb{R}^{B \times C \times D \times H \times W}$, where $D$ denotes the depth dimension. The Non-Subsampled Contourlet Transform (NSCT) is extended to 3D-NSCT, decomposing the volume into multi-scale, multi-directional subbands:

$$\{C_{l,\theta,\phi}\}_{l=1,\theta=1,\phi=1}^{L,\Theta_l,\Phi_l} = \mathcal{NSCT}^{3D}(I),  \qquad (24)$$

where $\Theta$ and $\Phi$ represent angular directions in 3D space. These subbands capture volumetric contour features, enhancing boundary awareness in 3D segmentation tasks.

### G.2   3D Gaussian Continuous Probability Field

The 2D Gaussian probability field is extended to 3D by defining a 3D Gaussian primitive at each voxel $(x, y, z)$:

$$g(\mathbf{p}) = \alpha \exp\left(-\frac{1}{2}(\mathbf{p} - \mu)^T \boldsymbol{\Sigma}^{-1}(\mathbf{p} - \mu)\right), \tag{25}$$

where $\mathbf{p} = [x, y, z]^T$, $\mu = [\mu_x, \mu_y, \mu_z]^T$ is the mean vector, $\boldsymbol{\Sigma}$ is a $3 \times 3$ covariance matrix, and $\alpha$ is the amplitude. The continuous Gaussian field is constructed as:

$$P(\mathbf{p}) = \sum_{i=1}^{N} w_i g_i(\mathbf{p}), \tag{26}$$

where $N = D \times H \times W$ is the number of voxels, and $w_i$ are learnable weights. The Contour-Guided Position Drift module adjusts the positions of the primitives based on 3D-NSCT subbands, ensuring precise boundary alignment in volumetric data. This formulation demonstrates GauSAM's theoretical compatibility with 3D tasks, leveraging the same principles of spatial continuity and local consistency to achieve robust segmentation performance.

### G.3   Adaptability to 3D Segmentation

The successful application of 2D Gaussian Splatting in GauSAM, combined with its contour-guided mechanism, provides a robust foundation for extending the framework to 3D scenarios. The 3D-NSCT enhances boundary fidelity through multi-scale, multi-directional structural priors, while the 3D Gaussian field ensures spatial continuity and local consistency in volumetric data. These adaptations enable GauSAM to model complex 3D medical imaging tasks effectively, facilitating the integration of multi-modal data and enhancing diagnostic precision across varied anatomical contexts.

## H   Further Discussion

### H.1   GauSAM's Strengths in Cross-Domain Generalization and Resolution-Agnostic Inference

GauSAM delivers exceptional performance in both binary polyp segmentation and multi-organ segmentation, excelling in cross-domain and cross-resolution scenarios, which achieves the highest Dice scores and ensures consistency. While GauSAM's source-domain performance on the BCV dataset achieves Dice scores comparable to state-of-the-art methods despite a higher HD, it demonstrates clear advantages in cross-domain scenarios—achieving significantly better Dice scores and lower HD compared to existing approaches. This superior generalization stems from GauSAM's continuous Gaussian probability field, which enables resolution-agnostic predictions and smooth modeling of large, uniform structures like the liver in AMOS dataset, enhancing boundary fidelity. The integration of contour-guided priors via the Non-Subsampled Contourlet Transform (NSCT) and Contour Propagation Unit (CPU) further boosts its robustness across domains. These strengths underscore the versatility and efficiency of GauSAM in medical image segmentation.

### H.2   Factors Impacting Multi-Organ Segmentation

The continuous representation that drives GauSAM's strengths also presents challenges for multi organ segmentation on the BCV dataset, influenced by its single feature and probability field design and dataset properties. GauSAM employs a single Gaussian probability field to represent all organ classes, which performs well for large structures but may have limited capacity to fully capture the fine details of smaller organs, given the diversity in size and boundary complexity of the BCV dataset. This design reflects a deliberate trade-off. While adopting multiple class-specific probability fields would significantly increase the number of trainable parameters to approximately 6.6M—substantially raising computational costs and inference latency—we found such overhead unacceptable. In contrast, GauSAM maintains a highly compact architecture with only 2.9M trainable parameters for multi

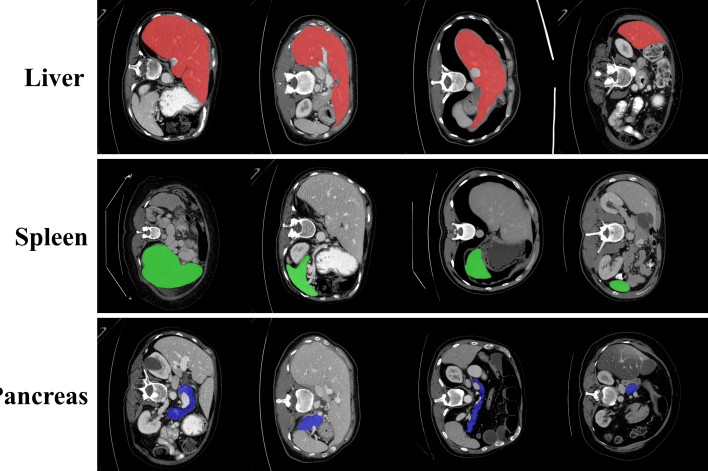

Figure 5: Visualization of 2D axial slices from the BCV dataset, highlighting appearance variability and semantic discontinuity across slices, which impacts consistent learning of organs.

organ segmentation, which is the lowest among models achieving comparable performance. We therefore prioritized efficiency and scalability, accepting a trade-off in fine-grained precision for smaller organs in multi-class settings.

Moreover, to adapt to 2D tasks, we converted the 3D CT volumes in the BCV dataset into axial slices. This slicing process introduces appearance variability across slices and disrupts semantic continuity, particularly affecting consistent learning of smaller organs due to the loss of 3D contextual information essential for volumetric coherence, as illustrated in Figure 5. However, GauSAM currently utilizes 2D Gaussian Splatting to construct continuous feature and probability fields for 2D images—a principle that naturally extends to 3D for reconstructing coherent anatomical structures. As a future direction, this extension can better capture semantic consistency and texture in CT scenarios, enabling more precise modeling while addressing these challenges. The ease of this 2D-to-3D transition further highlights GauSAM's strong extensibility and generalizability across diverse imaging tasks.

# I Broader Impact

## I.1 Clinical Impact

GauSAM holds significant promise for advancing medical image segmentation, with direct implications for clinical practice. By improving boundary fidelity and semantic consistency, GauSAM enables more precise structure delineation, facilitating accurate surgical planning and supporting radiologists in diagnosing complex cases. This can lead to earlier disease detection, improved patient outcomes, and reduced workload for medical professionals, particularly in medically underserved settings where automated tools are critical. Furthermore, such advancements may enhance the efficiency of treatment workflows and support telemedicine initiatives, broadening access to high-quality diagnostic tools in underserved regions.

## I.2 Resolution-Agnostic Mask Generation

The resolution-agnostic mask generation capability of GauSAM offers substantial practical benefits in medical imaging. Medical imaging modalities, such as CT and MRI, often produce high-resolution images due to the inherent properties of their imaging principles, which capture fine anatomical details. However, performing segmentation directly on these high-resolution images is computationally expensive, requiring significant resources. GauSAM's ability to generate accurate segmentation masks from downsampled images, while producing masks at the original image resolution with preserved quality, significantly reduces computational demands. This approach enables efficient processing on resource-constrained systems without compromising the precision of boundary delineation in the

final high-resolution output, making it highly applicable in clinical settings where computational efficiency is critical.

## I.3  Generalization Beyond Medical Imaging

Although GauSAM was trained and evaluated on medical image segmentation datasets, its design is not exclusively tailored to medical data. The challenges addressed by GauSAM—such as blurred boundaries, insufficient semantic consistency, segmentation artifacts, and inadequate boundary continuity and precision—are prevalent across various segmentation tasks in both medical and non-medical domains. These issues are particularly pronounced in medical image segmentation due to its complexity and significant clinical implications, making it a compelling evaluation setting for validating GauSAM's capabilities. However, the underlying principles of GauSAM, including its continuous Gaussian probability field and contour-guided mechanisms, are broadly applicable to other domains facing similar challenges, such as autonomous driving, remote sensing, or industrial defect detection. By improving segmentation accuracy, reducing artifacts, and enhancing boundary continuity, GauSAM has the potential to deliver strong performance in these fields, underscoring its versatility and robustness across diverse applications.

