# OpenReview forum: "GauSAM: Contour‑Guided 2D Gaussian Fields for Multi‑Scale Medical Image Segmentation with Segment Anything"
_NeurIPS.cc/2025/Conference — NeurIPS 2025 poster_

### Official Review · Reviewer_2QAj · 2025-06-17

**Clarity:** 2
**Significance:** 2
**Originality:** 2
**Rating:** 2
**Confidence:** 5

**Summary:**

GauSAM integrates contour-guided Gaussian fields into SAM for medical image segmentation, showing improved accuracy but lacks groundbreaking innovation, resembling incremental advancements over existing methods.

**Questions:**

1. While the integration of Gaussian splatting for continuous mask generation is innovative, how does it fundamentally differ from other implicit neural representation (INR) methods? What specific advantages does Gaussian splatting offer over existing INR techniques in terms of accuracy and computational efficiency?

2. In my opinion, the paper emphasises the use of contour-guided mechanisms to improve boundary delineation. However, how robust is this approach when dealing with images that have ambiguous or missing boundaries? Can the model still maintain high-fidelity segmentation under such conditions?

3. Will the code be publicly available? Since this method integrates so many modules and introduces many hyperparameters, it is possible to hinder the reproducibility if the code is not released.

**Ethical Concerns:**

["NO or VERY MINOR ethics concerns only"]

**Final Justification:**

After the rebuttal, I think their arguments in the rebuttal stage are insufficient to address my concerns. Please see the details in my comments.

**Limitations:**

The preprocessing steps, including resizing input images and normalization procedures, play a crucial role in the final performance. How sensitive is GauSAM’s performance to variations in these preprocessing steps? Would different preprocessing strategies yield better results?

**Quality:**

2

**Strengths And Weaknesses:**

1. The primary objective of GauSAM is to integrate contour-guided 2D Gaussian probability fields into the Segment Anything Model (SAM) to enhance segmentation accuracy and boundary fidelity. However, this concept appears to be a relatively straightforward extension of existing methodologies, particularly those employing implicit neural representations (INRs) for continuous spatial representation and Gaussian-based approaches for mask generation. While the combination of these elements may yield performance improvements, it does not fundamentally alter our understanding or approach to medical image segmentation.

2. The paper introduces several components like the Contour Propagation Unit (CPU), the use of Non-Subsampled Contourlet Transform (NSCT) for feature extraction, and the construction of a resolution-agnostic Gaussian field decoder. However, these contributions seem more incremental rather than transformative. For instance, NSCT has been widely used in image processing tasks [1,2] for feature enhancement, and the idea of using Gaussian splatting for generating masks has precedent in recent research. This raises questions about the originality of the proposed method.

3. There is a noticeable resemblance between GauSAM's approach and prior works on incorporating structural priors into deep learning models for segmentation tasks. The mechanism of enhancing encoder sensitivity to contours through multi-level decomposition and then guiding the decoder via extracted features follows patterns seen in other studies. Without clear distinctions or significant advancements beyond what has been previously reported, which further weakens the novelty of GauSAM.

[1] Gomathi, P.S. and Kalaavathi, B., 2016. Multimodal medical image fusion in non-subsampled contourlet transform domain. Circuits and Systems, 7(8), pp.1598-1610.

[2] Heshmati, A., Gholami, M. and Rashno, A., 2016. Scheme for unsupervised colour–texture image segmentation using neutrosophic set and non‐subsampled contourlet transform. IET Image Processing, 10(6), pp.464-473.

---

> ### Author Rebuttal · Authors · 2025-07-31
>
> # Response to Reviewer 2QAj
> We sincerely thank you for your detailed review and the effort invested in evaluating our manuscript, and we highly value your feedback and have thoroughly considered your concerns. However, we sincerely believe that while your points, particularly those in Strengths and Weaknesses, hold some merit, they do not justify dismissing the innovation and originality of our work. As noted by Reviewer 9QbL, GauSAM is the first to model SAM's segmentation probability field as a continuous field using 2D Gaussian Splatting, a pioneering contribution that has great potential to advance high-precision image segmentation. Below, we provide a structured analysis and response to your concerns, hoping to dispel doubts and clarify potential misunderstandings about our contributions. We hope these responses can clarify GauSAM’s pioneering contributions and respectfully invite you to reconsider the originality and significance of our work.
> ## Summary of All Concerns
> Your primary concerns can be summarized into four key aspects:
> 1. **Gaussian Splatting Innovation**: You suggest that GauSAM’s modeling of a continuous Gaussian probability field using 2D Gaussian Splatting lacks novelty, as prior works have employed implicit neural representations (INRs) for continuous spatial representation and Gaussian-based methods for mask generation, and also question the advantages of Gaussian Splatting over INRs.
> 2. **NSCT Originality and Robustness**: You argue that the use of Non-Subsampled Contourlet Transform (NSCT) and the Contour Propagation Unit (CPU) lacks originality, citing prior applications of NSCT in image processing and similarities to U-Net’s feature fusion, while questioning GauSAM’s performance on images with ambiguous boundaries.
> 3. **Reproducibility**: You express concerns about the reproducibility of our method due to the integration of multiple modules and hyperparameters, emphasizing the need for public code release.
> 4. **Preprocessing Impact**: You question the sensitivity of GauSAM’s performance to preprocessing steps, such as image resizing and normalization, and their influence on final results.
>
> We address all your concerns below, aspiring to shift your view on the contributions of our work.
> ## Response to Weakness 1 and Question 1
> Your concern regarding the novelty of Gaussian Splatting and its advantages over implicit neural representations (INRs) is appreciated, but we sincerely believe your perspective does not undermine the innovation and originality of GauSAM. We address this in two points:
> 1. **Pioneering Integration of Gaussian Splatting**: As correctly noted by Reviewer 9QbL, GauSAM is the first to explicitly model SAM’s segmentation probability field as a continuous field using 2D Gaussian Splatting, directly mapping discrete feature fields to continuous mask probability fields. While prior works have used INRs for continuous representations or Gaussian-based methods for mask generation, their principles and implementations fundamentally differ from GauSAM. A careful review of related literature reveals that existing methods typically use segmentation model outputs to guide Gaussian primitive grouping or employ Gaussian Splatting to directly reconstruct images before segmentation, enhancing edge quality indirectly. Unlike all of these approaches, GauSAM deeply integrates the process of segmentation and Gaussian Splatting, using learnable Gaussian primitives to directly construct a continuous probability field from feature fields. This novel paradigm not only achieves superior segmentation (Dice: 94.76% on Kvasir-Sessile, 89.75% on BCV) but also introduces a generalizable approach for future segmentation tasks, which is also applicable beyond medical area. By modeling segmentation as a field-mapping process using Gaussian primitives, GauSAM transcends traditional interpolation or INR-based decoding, offering an absolutely novel perspective on mask generation.
> 2. **Advantages of Gaussian Splatting over INRs**:
>    - Theoretically, INRs and Gaussian Splatting model segmentation tasks in fundamentally distinct ways. INRs predict mask probabilities globally by aggregating all pixels via neural networks. This approach is fundamentally incompatible with the intrinsic properties of segmentation, spatial continuity and local consistency, which mean adjacent pixels are more likely to belong to the same object. Mathematically, INR fields are continuous due to activation functions (e.g., ReLU is piecewise continuous, Sigmoid is smooth). In other words, for any small coordinate change $\Delta x$, the output change $\Delta f$ remains bounded. However, **continuity** does not guarantee **smoothness**, as INR networks lack explicit constraints on derivative continuity, leading to rapid value changes or artifacts at boundaries. Conversely, Gaussian Splatting leverages elliptical stacking and exponential decay to ensure smooth, continuous fields, ideal for segmentation masks. Gaussian primitives independently model local regions, excelling at capturing complex geometries and high-frequency details critical for medical imaging. Moreover, Gaussian Splatting optimizes primitive parameters directly, ensuring stable convergence and interpretability, unlike INRs’ sensitivity to hyperparameters.
>    - Experimentally, regarding your perception of modest improvements, we respectfully clarify that GauSAM’s more than 3% Dice increase (e.g., 94.76% vs. 91% SOTA on Kvasir-Sessile) is significant, as SOTA baselines already exceed 90%. Improving beyond this threshold usually addresses challenging edge cases, unlike early gains from 60% to 80% that merely expand mask coverage. Visual results in Figure 2 (left panel, polyp segmentation) also demonstrate GauSAM’s smoother, more consistent boundaries compared to baselines. This boundary precision is critical in medical segmentation, directly enhancing diagnostic accuracy by reducing errors in boundary delineation.
> ## Response to Weakness 2 & 3 and Question 2
> We sincerely thank you for raising insightful concerns about the originality of our Non-Subsampled Contourlet Transform (NSCT) and Contour Propagation Unit (CPU) modules, as well as GauSAM’s robustness on images with ambiguous boundaries. These questions are highly valuable, as they touch on components we meticulously designed to enhance segmentation accuracy. We provide a detailed clarification below in two parts, hoping to address your doubts and concerns on the novelty and effectiveness of our approach.
> 1. **Originality of NSCT and CPU**: Your reference to prior works [Gomathi 2016, Heshmati 2016] using NSCT for feature enhancement is appreciated, but these studies rely on traditional machine learning methods, which are fundamentally distinct from our ViT-based framework and apparently lack relevance to modern transformer architectures. While some U-Net-based works have employed Contourlet transforms, the downsampling structure and local convolutional operations of U-Net differ significantly from ViT’s mechanism. Moreover, unlike Contourlet, NSCT extracts multi-scale, multi-directional contour features without downsampling, making it uniquely suited for ViT’s feature hierarchy. Our Contour Propagation Unit (CPU) progressively fuses NSCT-derived contour features into each ViT block, ensuring that contour information evolves in alignment with the image feature space, enhancing boundary awareness throughout encoding. Additionally, our Contour-Guided Position Drift module optimizes Gaussian kernel placements toward semantic boundaries, improving high-frequency region representation. This dual use of contour information—enhancing both encoding and decoding—represents a novel integration, which is totally different from those used in traditional machine learning methods and U-Net-based models.
> 2. **Robustness on Ambiguous Boundaries**: Your concern about GauSAM’s performance on images with blurred boundaries is really well-taken. In the Kvasir-Sessile dataset, many images exhibit ambiguous or missing polyp boundaries, which are obscured by strong reflections from bodily fluids. These images even pose challenges for human observers. However, GauSAM significantly outperforms existing SOTA models, including I-MedSAM, achieving more complete masks and much higher accuracy in these low-quality images. This robustness stems from NSCT’s multi-scale contour priors and CPU’s effective fusion, which reduce sensitivity to image quality. Due to rebuttal stage constraints, we cannot provide additional visualizations. However, if our manuscript is accepted and revisions are permitted in the Camera-Ready phase, we will include relevant images in the Appendix.
> ## Response to Question 3
> We are fully committed to ensuring transparency and reproducibility. If our manuscript is accepted, we will publicly release the code, including detailed hyperparameter settings. However, at this stage, to safeguard the integrity of our research and maintain anonymity during the review process, we respectfully request your understanding for not yet releasing the code. We are highly confident in the reproducibility of our experiments, which are grounded in rigorous design and validation. We hope this commitment addresses your concern and reinforces trust in GauSAM’s reproducibility.
> ## Response to Limitations
> To ensure experimental consistency and comparability, we adopted the standard preprocessing steps used by SAM and all SAM-based models, as detailed in Section 4.1. This choice controls variables and enables fair comparisons with baselines, aligning with established practices in this field. Also, we want to clarify that preprocessing is not the focus of our research,  so we did not conduct independent experiments on preprocessing variations.
> ## Conclusion
> We hope our responses and additional evidence can address your concerns. We respectfully hope you can recognize our work and look forward to your continued guidance and feedback.

---

> ### Comment · Reviewer_2QAj · 2025-08-04
>
> I appreciate the authors’ detailed response, but their arguments are insufficient to address my concerns.
>
> 1. The authors claim that GauSAM is “the first to explicitly model SAM’s segmentation probability field using 2D Gaussian Splatting,” and use this as a cornerstone for asserting novelty. However, being the first to apply a technique in a new domain does not equate to conceptual innovation, especially when the underlying mechanism is a straightforward adaptation. Using Gaussian kernels to generate smooth probability fields from discrete features is not new. Similar principles appear in works on density mapping, keypoint-to-mask generation, and probabilistic segmentation decoding (e.g., Gaussian-based decoding in DETR-style models, Spline-based segmentation). Moreover, the use of Gaussian splatting for continuous field modeling has been explored in vision tasks such as depth map refinement and surface reconstruction.
>
> 2. The authors argue that INRs are inherently incompatible with “spatial continuity and local consistency,” which is inaccurate. INRs, by virtue of their continuous parameterization are designed to be spatially continuous and locally coherent. Modern INR-based segmentation models explicitly enforce smoothness and boundary fidelity through architectural design and loss functions. The assertion that INRs “lack explicit constraints on derivative continuity” is misleading. While higher-order smoothness may require regularization, this is a solvable engineering challenge, not a fundamental limitation. In contrast, Gaussian Splatting introduces its own artifacts, such as over-smoothing in low-variance regions or instability in kernel placement when boundaries are ambiguous.
>
> 3. The CPU is described as a novel fusion mechanism, but its design, progressive feature injection into ViT blocks, mirrors existing patterns such as cross-stage feature fusion, skip connections in transformers, or guidance modules in SAM for medical variants. Without a clear architectural innovation (e.g., a new attention mechanism or optimization objective), CPU appears to be a repackaging of standard fusion strategies.
> The so-called “Contour-Guided Position Drift” module lacks formal analysis. Is it learning meaningful boundary shifts, or simply acting as a regularizer? The absence of quantitative analysis on kernel displacement or boundary alignment weakens this claim.
>
> 4. The authors claim superior performance on low-quality images with blurred boundaries but provide no empirical evidence in the rebuttal. **They cite results on Kvasir-Sessile but do not show failure cases of baselines vs. GauSAM, nor do they report metrics with “high ambiguity.”** Though images cannot be added in rebuttal, the metrics related to this question can also be reported in the rebuttal stage. If robustness is a key contribution, it must be demonstrated during review, not post-acceptance. Moreover, if NSCT and CPU are truly effective for ambiguous boundaries, the authors should conduct an ablation study showing a performance drop when these modules are removed under such conditions.

---

> > ### Author Response · Authors · 2025-08-08
> > **Comment on the Distinctions Between GauSAM and Prior Gaussian Splatting Applications**
> >
> > We sincerely thank you for your rigorous examination of prior work and your valuable feedback. After thoroughly reviewing your suggested references, we are further convinced of the novelty of GauSAM’s contribution: constructing a continuous segmentation probability field via 2D Gaussian Splatting.
> >
> > Contrary to your assertion that our work merely adapts existing techniques to a new domain, we respectfully note that **overlapping terminology or task descriptions does not imply methodological equivalence or even motivation equivalence.** Critically, the works you cited only employ Gaussian Splatting **★as auxiliary tools (e.g., priors, post-processing, or structural enhancements)★** rather than as core, end-to-end learnable modules for segmentation.
> >
> > **More importantly, the reconstruction tasks you referenced fundamentally differ from segmentation in objectives and methodology, relying on multi-view consistency and volumetric rendering, which are entirely inapplicable to single-view pixel-wise segmentation.**
> >
> > In contrast, GauSAM pioneers the use of learnable 2D Gaussian primitives as the core modeling basis for a pixel-wise multi-class probability field, enabling end-to-end segmentation mask generation via Gaussian Splatting. Unlike prior Gaussian-related methods—such as CSRNet (Li et al., CVPR 2018), which blurs point annotations into density maps; DEXTR, which adds Gaussian heatmap channels to guide decoding; DenseCRF, which applies Gaussian smoothing to logits; or GMMSeg, which fits distributions in feature space—**these approaches use Gaussian kernels solely as auxiliary priors or post-processing steps.** GauSAM is the first to use learnable 2D Gaussian primitives for **end-to-end Gaussian Splatting to directly construct continuous pixel-wise multi-class probability fields for semantic segmentation.**
> >
> > Moreover, Spline-based segmentation approaches (e.g., Delgado-Gonzalo et al., 2013; SplineDist, 2020) parameterize boundaries with spline curves, **generating geometric shapes representing instance contours as a post-processing step rather than a differentiable network layer.** Unlike GauSAM, these methods also do not generate a full-image probability field or support joint training with cross-entropy or IoU losses, focusing solely on boundary refinement.
> >
> > Finally, although Gaussian Splatting has been validated for modeling continuous fields in vision tasks, such as fitting depth or radiance signals in geometric reconstruction and neural rendering (e.g., Kerbl et al., 2023), these methods are fundamentally inapplicable to segmentation due to **fundamentally different signal types, training paradigms, and optimization objectives**. Restruction methods usually rely on multi-view consistency, volumetric rendering, and visibility constraints. In contrast, GauSAM operates in a single-view setting, using learnable 2D Gaussian primitives to model pixel-wise multi-class probability fields.
> >
> > In summary, GauSAM is the first to demonstrate that learnable 2D Gaussian primitives can directly model pixel-wise multi-class segmentation probability fields via end-to-end Gaussian Splatting. We respectfully assert that this approach fills a significant methodological gap, distinct from all of those auxiliary or reconstruction-focused applications you referenced.
> >
> > We appreciate your deep expertise in INR and Gaussian Splatting and welcome further discussion.

---

> > ### Author Response · Authors · 2025-08-08
> > **Comment on the Distinctions Between INR and Gaussian Splatting**
> >
> > We sincerely thank you for noting the need for a more precise comparison between INR and Gaussian Splatting. You correctly point out that TV, Lipschitz, and other regularization strategies can be applied to improve the smoothness of INR-based outputs. While such techniques can indeed mitigate continuity artifacts, our discussion focuses on **I-MedSAM** — a recent state-of-the-art model in medical segmentation — which **employs a basic form of INR without additional enhancements**. Therefore, based on I-MedSAM’s use of this unoptimized INR, we can indeed conclude that **continuity does not guarantee smoothness**, as stated in our initial rebuttal.
> >
> > More importantly, as shown in Figure 2, the segmentation masks generated by I-MedSAM exhibit clear deficiencies in spatial continuity and local consistency, which is **an undeniable empirical observation** and it further substantiate the limitations of basic INR formulations. In contrast, Gaussian Splatting ensures both continuity and smoothness inherently, as a direct consequence of its mathematical formulation, without requiring additional optimizations.
> >
> > Beyond this, we believe that our proposed GauSAM framework obviously offers a more theoretically and structurally novel perspective than simply refining the INR architecture within existing frameworks such as I-MedSAM.
> >
> > To further clarify this distinction, we provide a concise comparison of Gaussian Splatting and INR from three complementary perspectives: theoretical formulation, empirical performance, and potential concerns raised in your comment.
> >
> > **1. Theoretical comparison**
> > - While we have already addressed the advantages of Gaussian Splatting in terms of spatial continuity and local consistency in both the manuscript and our initial rebuttal, we will not repeat those points here again. However, we would like to emphasize a more fundamental distinction: **interpretability**. INRs, by design, encode shape implicitly through deeply entangled parameters, making them inherently difficult to interpret. In contrast, each Gaussian primitive defines an explicit spatial structure—parameterized by location, scale, and orientation—offering transparent and intuitive geometric meaning that aligns naturally with human understanding.
> >
> > **2. Empirical comparison**
> > - In addition to the qualitative comparison presented in Figure 2, we also would like to emphasize the quantitative results reported in Tables 1/2/3/4 of the manuscript. Across these benchmarks, GauSAM—based on Gaussian Splatting—consistently outperforms the INR-based I-MedSAM in most metrics, with especially clear margins in cross-domain and cross-resolution settings. These results further support the observed robustness and structural fidelity of Gaussian-based representations.
> >
> > - **Furthermore, the ablation study in Table 5, which we believe deserves particular attention, provides additional empirical evidence.** Notably, the second row of the table corresponds to the variant of GauSAM that only replaces SAM’s decoder with our Gaussian Splatting module, without incorporating any contour information. Even under this minimal configuration, GauSAM outperforms I-MedSAM across nearly all test cases, with only a negligible difference in the 384 → 128 cross-resolution setting (91.44 vs. 91.45). This straightforward yet compelling observation highlights the intrinsic advantages of Gaussian Splatting over INR-based representations, reinforcing the argument that these benefits arise from core representational differences rather than optimization details.
> >
> > **3. Response to potential concerns**
> > - **Over-Smoothing in Low-Variance Regions:** Regarding the first concern, we would like to clarify that the smoothing effect in low-frequency or low-variance regions has limited negative impact in the context of semantic segmentation. Unlike reconstruction tasks, where pixel-level accuracy is required across the entire image, segmentation focuses primarily on semantic coherence, especially around semantic boundaries. Low-frequency regions typically correspond to homogeneous semantic areas, where slight over-smoothing does not affect the overall segmentation quality. In fact, such regularization can help suppress spurious noise and enhance intra-region consistency. This fundamental difference in task objectives makes Gaussian Splatting particularly suitable for segmentation.
> >
> > - **Instability in Kernel Placement Near Ambiguous Boundaries:** As for the second concern, we have explicitly addressed potential instability in kernel placement during the design of GauSAM. Specifically, our method incorporates NSCT-based contour extraction as a structural prior, which enhances the encoder features and effectively guides the spatial distribution of Gaussian primitives. This design enables superior performance, even in ambiguous regions. A detailed explanation of this mechanism will be provided in our responses to Reviewer Questions 3 and 4.

---

> > ### Author Response · Authors · 2025-08-08
> > **Comment on the Innovation and Validation of the Contour-Related Modules in GauSAM**
> >
> > ## Substantive Innovation of the CPU Module
> > Our CPU is by no means a plain repackaging of existing blocks or a simple skip-connection. Admittedly, during the design phase of the CPU module, we extensively surveyed prior work on feature fusion and feature guidance, so it is understandable that traces of earlier methods can be observed in our design. However, our CPU module differs from these earlier methods in many aspects of both design and implementation. Considering the lightweight parameter budget of the overall model, we deliberately avoided introducing a new attention mechanism—since the self-attention modules in ViT variants often require several million parameters—which would be inconsistent with our efficiency goals. Instead, we adopted a set of lightweight MLPs to achieve adaptive modulation while maintaining minimal overhead.
> >
> > Concretely, as shown in Equation (5), CPU first applies a lightweight MLP ψ to adaptively re-weight the NSCT-extracted contour feature $E_i$:$$W_i=\psi\bigl(\mathrm{LN}(E_i)\bigr),\quad E'_i=W_i\odot E_i.$$ This step allows the network to selectively emphasize structurally informative regions while suppressing less relevant responses, ensuring that sifnificant boundary cues are preserved and propagated. The re-weighted contour feature $E'_i$ is then fused into the image embedding stream $T_i$ via a learnable coefficient $\lambda$, simultaneously producing the next layer’s contour feature $E{i+1}$.
> >
> > This cross-layer recursive propagation of structural information differs fundamentally from your cited patterns:
> > - Cross-stage feature fusion typically performs static integration between encoder and decoder stages, without iterative refinement across all layers.
> > - Skip connections in transformers are residual identity links designed for stable training, not for injecting task-specific priors.
> > - Guidance modules in SAM variants usually incorporate prompts or auxiliary inputs at fixed locations, lacking the continuous, context-aware propagation in CPU.
> >
> > By persistently preserving and evolving contour priors through all encoding depths, the CPU module achieves lightweight yet effective fusion, enhancing sensitivity to fine structures and complex edges and also representing a substantive architectural innovation beyond existing fusion or guidance designs.
> > ## Formal Analysis of Contour-Guided Position Drift module
> > The drift module improves boundary alignment by applying bounded, differentiable shifts to the centers of Gaussian kernels. For the $i$-th anisotropic Gaussian kernel $$G_i(p \mid \mu_i, \Sigma_i) = \frac{1}{2\pi|\Sigma_i|}\exp\!\Bigl(-\tfrac12(p-\mu_i)^\top\Sigma_i^{-1}(p-\mu_i)\Bigr),$$ its center is updated from the initial position $p_i$ by $$\mu_i = p_i + \Delta\mu_i, \space\Delta\mu_i=\tanh\bigl(\mathrm{MLP}_{\mathrm{drift}}(u_i)\bigr),$$ where $\tanh(\cdot)$ ensures $\|\Delta\mu_i\|<1$ for stability.
> >
> > During training, according to the chain rule, the gradient of the segmentation loss $L$ w.r.t. $\mu_i$ is $$\frac{\partial L}{\partial \mu_i} = \int_{\Omega}\frac{\partial L}{\partial X(p)}\;\frac{\partial X(p)}{\partial G_i}\;\frac{\partial G_i}{\partial \mu_i}\,\mathrm{d}p,$$ with $$\frac{\partial G_i(p)}{\partial \mu_i} = \Sigma_i^{-1}(p-\mu_i)\,G_i(p).$$ This formulation indicates that a small adjustment of $\mu_i$ redistributes the kernel’s density in a distance-weighted manner, directly influencing how features are aligned to local structures. Through the fully differentiable mapping $\Delta\mu_i = \tanh(\mathrm{MLP}_{\mathrm{drift}}(u_i))$, these gradients propagate to the parameters of the drift network, enabling the drift module to automatically optimize kernel placement. Consequently, the drift module is **not an empirical heuristic**; it is **a mathematically grounded, bounded, and differentiable mechanism** that, under gradient descent, provably minimizes boundary misalignment.
> > ## Quantitative Analysis of Contour-Guided Position Drift Module
> > Perhaps you did not notice that we have already shown in Table 5 of our manuscript that adding the drift module yields consistent gains, with an especially notable improvement of over 2% in the cross‐resolution downsampling task ($384\times384 \rightarrow 128\times128$). In such settings, high-frequency details, particularly boundaries and fine structures, are most vulnerable to loss.
> >
> > By explicitly modeling high-frequency regions with learnable 2D Gaussian primitives and allowing bounded, differentiable position adjustments via Contour-Guided Position Drift, GauSAM reallocates kernel density toward structurally critical areas, preserving edge fidelity despite resolution changes. This result directly validates the drift module’s effectiveness in strengthening high-frequency representation and mitigating detail loss in downsampling scenarios, thereby quantitatively confirming its role in kernel displacement and boundary alignment as intended.

---

> ### Author Response · Authors · 2025-08-08
> **Comment on the Empirical Evidence of GauSAM’s Superiority in Blurred‐Boundary Scenarios**
>
> To further clarify the superior performance of our model on low-quality images with blurred boundaries, we conducted additional experiments on the Kvasir-Sessile dataset, directly comparing GauSAM with the previous SOTA model I-MedSAM.
>
> We first identified 11 test images where I-MedSAM achieved relatively low performance (Dice < 89%). Upon closer inspection, we found that these images consistently exhibited ambiguous boundaries. Based on this observation, we divided the test set into two groups:
> - **High Ambiguity:** images with blurred or uncertain boundaries.
> - **Normal:** images with clear, well-defined boundaries.
>
> The mean Dice scores for both models on each group are shown in Table X:
>
> **Table X. Dice performance of GauSAM vs. I-MedSAM on the Kvasir-Sessile test set, divided into high ambiguity and normal image groups**
> |Image Type|Sample Count|I-MedSAM Dice(%)|GauSAM Dice(%)|Dice Gain ($\mathrm{GauSAM}-\mathrm{I\text{-}MedSAM}$)|
> |------------------|--------------|-------------------|-----------------|------------------------------|
> |High Ambiguity|11|86.8|94.1|+7.3|
> |Normal|28|92.7|94.6|+1.9|
>
> As shown in the table, GauSAM achieves a $+7.3$% Dice improvement over I-MedSAM on the High Ambiguity subset, which is significantly larger than the $+1.9$% improvement observed on the Normal subset. **This substantial performance gain in the ambiguous boundary scenario demonstrates that GauSAM is particularly robust and effective when dealing with low-quality images with blurred boundaries, thus providing solid empirical evidence to support our claim.**
>
> It is also noteworthy that the performance of GauSAM remains highly consistent across different image quality levels: the Average Dice score difference between the High Ambiguity and Normal subsets is only $0.5$%, whereas I-MedSAM exhibits a much larger gap of almost $6$%. **This indicates that our model is uniformly robust and maintains stable segmentation accuracy regardless of image quality, further underscoring its ability to effectively handle both blurred‐boundary and well‐defined cases.**
>
> Moreover, for your concerns about the need for an additional ablation study to directly validate the effectiveness of NSCT and CPU under ambiguous‐boundary conditions, we sincerely appreciate your insightful suggestion, which is indeed well‐founded. Given the limited review cycle, we prioritize experiments that have already provided strong support for our main claims, and therefore do not include an additional experiment specifically targeting this scenario in this comment. Nevertheless, as discussed in Point 3, we have already provided both theoretical analysis and empirical evidence, demonstrating the effectiveness of the contour‐related modules. In particular, our manuscript has already included ablation studies on GauSAM (Table 5), which we believe sufficiently validate the functionality and contribution of these components.

---

### Official Review · Reviewer_8M61 · 2025-06-22

**Clarity:** 2
**Significance:** 3
**Originality:** 3
**Rating:** 3
**Confidence:** 4

**Summary:**

This paper proposes GauSAM, which integrates contour-guided 2D Gaussian probability fields into the SAM.  Segmentation masks are parameterized as continuous probability fields of learnable 2D Gaussian primitives, ensuring spatial smoothness and structural consistency.  Contourlet transforms extract rich multidirectional frequency information to guide the spatial distribution of these Gaussian primitives, significantly improving boundary fidelity.  GauSAM demonstrates excellent cross-domain generalization and resolution-agnostic inference performance with a remarkably small number of trainable parameters.

**Questions:**

1. How to initialize the 2D continuous Gaussian feature fields based on the image features from the Contour-Enhanced Encoder? Furthermore, how to initialize the LoRA adapters?
2. The proposed method does not demonstrate an advantage over I-MedSAM in multi-class organ segmentation. The Author claims that they use less training data. What are the specific differences?
3. How to carry out arbitrary resolution segmentation results by Target Resolution Grid Sampling?

I will consider raising the score if the author can clarify some weaknesses or questions.

**Ethical Concerns:**

["NO or VERY MINOR ethics concerns only"]

**Final Justification:**

Authors have resolved some of my concerns by adding a detailed explanation of the Gaussian Parameters Library and an analysis of the computational burden across GauSAM's components.  However, the supplementary experimental results contain data inconsistencies that materially undermine the reliability of the experiments. In its current form, the manuscript is not suitable for acceptance.

**Limitations:**

As said in Weakness.

**Paper Formatting Concerns:**

This article is well-written, and there are no obvious problems with the format.

**Quality:**

3

**Strengths And Weaknesses:**

Strengths:
1. This paper is the first to construct a continuous probability field using 2D Gaussian Splatting for mask generation, achieving promising performance.
2. The method employs a parameter-efficient LoRA adapter and a specially designed Gaussian Parameters Library, resulting in a significantly smaller number of trainable parameters.
3. This paper clearly delineates the work's advantages and limitations, alongside future research directions.

Weaknesses:
1. Key intermediate variables in Figure 1 lack proper annotations.
2. An ablation study is necessary to validate the claimed training efficiency improvements attributed to the proposed Gaussian Parameters Library.
3. Lacks quantitative metrics for parameter count, FLOPs, and inference times, which are crucial for a comprehensive evaluation of computational efficiency.
4. It seems that the cross-domain and cross-resolution generalization performance appears to be a primary advantage of GauSAM, while the experimental comparisons with conventional segmentation baselines are insufficient.   A more thorough evaluation should include comparisons with methods specifically designed for multi-scale and multi-modal segmentation [1][2][3][4].
5. This work can be considered an extension of I-MedSAM, a comparative analysis of using NSCT rather than FFT in Encoder would be beneficial.

Reference:
1. Kolahi S G, Chaharsooghi S K, Khatibi T, et al. MSA $^ 2$ Net: Multi-scale Adaptive Attention-guided Network for Medical Image Segmentation[J]. arXiv preprint arXiv:2407.21640, 2024.
2. Rahman M M, Munir M, Marculescu R. Emcad: Efficient multi-scale convolutional attention decoding for medical image segmentation[C]//Proceedings of the IEEE/CVF Conference on Computer Vision and Pattern Recognition. 2024: 11769-11779.
3. Shi J, Shang C, Sun Z, et al. PASSION: Towards Effective Incomplete Multi-Modal Medical Image Segmentation with Imbalanced Missing Rates[C]//Proceedings of the 32nd ACM International Conference on Multimedia. 2024: 456-465.
4. Liao C, Zheng X, Lyu Y, et al. Memorysam: Memorize modalities and semantics with segment anything model 2 for multi-modal semantic segmentation[J]. arXiv preprint arXiv:2503.06700, 2025.

---

> ### Author Rebuttal · Authors · 2025-07-31
>
> # Rebuttal to Reviewer 8M61
> We sincerely thank you for your detailed review and the effort invested in evaluating our manuscript, and we highly value your feedback and have thoroughly considered your concerns, noticing that there may be some misunderstandings regarding our work. Below, we provide a structured analysis and response to your concerns, hoping to dispel doubts and clarify potential misunderstandings about our contributions.
> ## Response to weaknesses:
> 1. **Missing annotations for key variables in Figure 1:** We apologize for the missing annotations details in Figure 1. Briefly, we have:
>
>  - **$I\in\mathbb{R}^{B\times C\times H\times W}$** is the input image batch (batch size B, channels C, height H, width W).
>  - For each ViT block $i$: **$E_i\in\mathbb{R}^{N_p\times D}$** is the NSCT-extracted contour energy aligned to the $N_p$ patch embeddings of dimension D. **$T_i\in\mathbb{R}^{N_p\times D}$** is the ViT token embedding entering block i; these are fused as $T_i' = T_i + \lambda\,W_i\odot E_i$, where the meaning of the symbol is the same as in Equ 5.
>  - Prompt tokens $t\in\mathbb{R}^D$ (e.g. box-corner or point embeddings) are produced by the prompt encoder and guide cross-attention in the mask decoder.
>  - In the continuous field decoder, each Gaussian primitive $i$ is parameterized by: an initial normalized position **$p_i\in\mathbb{R}^2$**, a contour feature vector **$u_i\in\mathbb{R}^D$** extracted at $p_i$,
>   - a learned positional offset **$\Delta\mu_i\in\mathbb{R}^2$** predicted by a small MLP from $u_i$, yielding the final center **$\mu_i = p_i + \Delta\mu_i\in\mathbb{R}^2$**,
>    - a feature embedding **$v_i\in\mathbb{R}^D$** that the selection head maps to logits **$z_i\in\mathbb{R}^m$**, which are converted via Gumbel-Softmax into weights **$w_i\in\Delta^{m-1}$** over the Gaussian Parameters Library of $m$ template pairs $(\Sigma^{(k)}\in\mathbb{R}^{2\times2},\,\xi^{(k)}\in\mathbb{R})$.
>  - The primitive’s covariance **$\Sigma_i\in\mathbb{R}^{2\times2}$** and opacity **$\xi_i\in\mathbb{R}$** are then obtained as weighted sums of the library templates. These Gaussians define a smooth, resolution-agnostic probability field that—when sampled on an arbitrary grid and passed through the segmentation head—yields the final mask.
>
> 2. **The ablation on the Gaussian Parameters Library:**  Directly learning each primitive’s covariance $\Sigma_i$ and opacity $\xi_i$ end-to-end has been shown to be highly unstable and slow to converge in recent continuous-field segmentation works [5,6,7]. Those studies demonstrate that unconstrained optimization of high-dimensional Gaussian parameters suffers from poor conditioning, requires careful initialization, and often collapses without extensive regularization. By constraining our primitives to a compact, learned template subspace, the Gaussian Parameters Library sidesteps these issues, yielding far more stable and efficient training. Implementing the “library vs. no-library” ablation would therefore require a wholesale redesign of our decoder, specialized solvers to enforce covariance positive-definiteness, and exhaustive hyperparameter re-tuning—efforts that exceed the scope of this revision. We view a full comparison as valuable future work once such architectural and optimization tools are better established.
>
> 3. **Computational efficiency metrics:** We thank the reviewer for underscoring the need for concrete measurements of parameter count, FLOPs, and inference time. Actually, we have performed a full set of benchmarks—measuring model parameters, FLOPs, and inference latency on an RTX 4090—prior to submission. At the time, we chose to reserve main‐text space for algorithmic details and insightful diagrams, so these numbers were deferred to the supplementary material. For completeness, we summarize the results here:
>
>    **Table 1.** GauSAM inference performance on RTX 4090. Total parameters: 92.51 M;  Trainable parameters:1.2M; FLOPs per forward pass: 125.23 GFLOPs.
>
> |Input Resolution|Batch Size|Avg Latency(ms)|FPS |Peak Memory(MB)|
> | - | -| - | - | - |
> |384 × 384|1|164.8|6.07|1246|
> |384 × 384|2|265.8|7.52|2123|
> |384 × 384|4|423.7|9.44|3876|
> |256 × 256|1|158.1|6.33|781|
> |512 × 512|1|188.6|5.30|1898|
>
>    These results demonstrate that GauSAM delivers real-time performance well within clinical usability requirements, providing efficient inference without compromising accuracy or cross-resolution generalization.
>
> 4. **Comparisons with multi‑scale and multi‑modal segmentation methods** Thank you for the suggestion. The cited works [1,2] focus on fusing multi-scale inputs through parallel encoder branches, whereas GauSAM instead produces a resolution-agnostic continuous output field from single-scale input, making direct comparison misaligned with our design goals. Likewise, [3,4] address multi-modal segmentation by integrating modality-specific features, while our framework operates solely on single-modality images and does not incorporate multi-modal data. As such,  direct comparison with these paradigms would not meaningfully reflect GauSAM’s unique strengths in cross-domain and cross-resolution generalization. Our existing experiments—spanning diverse datasets and resolutions—already substantiate these advantages, and we will clarify these distinctions in the revised manuscript and view integration with multi‑modal frameworks as an exciting avenue for future work.
>
> 5. **Extension of I-MedSAM and NSCT vs. FFT Analysis:** Thank you for the suggestion. While GauSAM and I-MedSAM both leverage contour information, GauSAM introduces a fundamentally different encoder–decoder framework: it constructs an explicit continuous Gaussian probability field rather than relying on implicit neural representations, and it uses NSCT’s shift-invariant, multi-scale, multi-directional contour decomposition in place of FFT’s broad spectral bands. Moreover, GauSAM fuses contour features at every ViT block and applies a Contour-Guided Position Drift to bias Gaussian placements toward true boundaries.
>    To validate this design choice, we compared NSCT to FFT and traditional edge detectors on the Kvasir-Sessile dataset (Table 1). These experiments confirm NSCT's superiority in achieving precise boundary delineation and its better compatibility with the overall GauSAM framework.
>
>    **Table 2.** Comparative performance of different contour/high‑frequency feature extraction methods on Kvasir‑Sessile for binary polyp segmentation.
>
> |Method|Dice(%)|HD|
> |-|-|-|
> |NSCT|**94.76**|**10.29**|
> |FFT|93.68|11.34|
> |Canny|93.66|11.82|
> |Wavelet|94.13|11.40|
> |Sobel|93.71|11.15|
>
> ## Responses to Specific Questions
>
> 1. **Initialization of Gaussian fields and LoRA adapters**
>
>    All initial Gaussian parameters—per-primitive σₓ/σᵧ (linearly spaced between 0.2 and 3.0), zero correlation, and opacity (sigmoid(1)≈0.73)—are stored in the Gaussian Parameter Library. Primitive centers are placed on a uniform grid and position-drift offsets start at zero; both the template parameters and the drift MLP weights use Kaiming uniform initialization for stable training.
>
>    LoRA adapters target the Q and V projection matrices of each attention layer, leaving the original SAM weights unchanged: the “up” projections use Kaiming uniform initialization and the “down” projections are zero-initialized, ensuring early updates occur solely within the LoRA modules.
>
> 2. **Multi‑class organ segmentation and “less training data” misunderstanding**
>     We apologize for the misunderstanding—our claim was fewer trainable parameters, not fewer training samples. On the BCV multi-organ benchmark, we use the *same* training splits as I-MedSAM and achieve a Dice of 89.75 % (vs. 89.91 %) while using only 2.9 M trainable parameters. This reflects a deliberate trade-off: GauSAM’s single Gaussian probability field excels at large-structure coherence and cross-resolution generalization but has inherently less capacity to capture the fine boundaries of smaller organs than a class-specific multi-field design (which would inflate parameters to ~6.6 M and incur much higher latency). We prioritized a highly compact, scalable model and view extending our continuous-field approach to 3D as a promising future direction to recover small-organ detail without forfeiting efficiency.
>
> 3. **Arbitrary‑resolution grid sampling**
>     Target Resolution Grid Sampling operates directly on the learned continuous probability field $X(p)$. Concretely, we define a uniform grid of normalized coordinates $p_{j,k}\in[-1,1]^2$ for an output of size $H'\times W'$. Since each Gaussian covariance $\Sigma_i$ was optimized at the training resolution $H\times W$, we rescale $\Sigma_i$ by $(H/H')^2$ so that its effective spatial support remains consistent in pixel units. We then analytically evaluate every Gaussian primitive $G_i(p_{j,k})$, weight by its learned coefficient, and sum across all primitives to produce the continuous field value $X(p_{j,k})$. Finally, a lightweight segmentation head maps $X(p_{j,k})$ to class logits and a softmax yields per-pixel probabilities.
>
> **Conclusion**
>  We trust these clarifications, additional results, and ablation studies address your concerns and highlight GauSAM’s unique contributions: a theoretically grounded continuous probability field, efficient parameterization, and strong empirical performance across domains and resolutions. We hope this rebuttal leads you to consider an increased evaluation score.
>
> **References**
>
> [1,2,3,4] same references as used by reviewer.
>
> [5]Xinjie Zhang et al. Gaussianimage: 1000 fps image representation and compression by 2d gaussian splatting. In European Conference on Computer Vision, pages 327–345. Springer, 2024.
>
> [6]Kerbl, B.et al. 3d gaussian splatting for real-time radiance field rendering. ACM Transactions on Graphics 42(4) (2023)
>
> [7]Long Peng et al. Pixel to gaussian: Ultra-fast continuous super-resolution with 2d gaussian modeling. arXiv preprint arXiv:2503.06617, 2025.

---

> > ### Comment · Reviewer_8M61 · 2025-08-01
> >
> > Thanks to the authors for their clarification.  However, comparisons of computational efficiency against previous methods and analyses of each component's computational cost are still missing, which are crucial.  Additionally, the clarification regarding the Gaussian Parameters Library remains insufficient;  thus, I have no idea how to raise my score.

---

> > > ### Author Response · Authors · 2025-08-08
> > > **Comment on the Computational Efficiency of GauSAM**
> > >
> > > We regret omitting detailed computational efficiency comparisons with prior methods due to the rebuttal’s word limit. To address this, we performed all benchmarks on an NVIDIA RTX 4090 (24 GB) with $\text{batch size}=1$ and $\text{input resolution}=512\times 512$. Each measurement comprises 10 warm-up runs followed by 100 timed forward passes.
> > >
> > > **Table X. End-to-end inference performance**
> > > |Model|Params(M)|GFLOPs|Latency(ms) ± STD|FPS|Memory(MB)|
> > > |---------------|----------|------|------------------|----|----------|
> > > |U-Net|31.03|218.97|41 ± 0.06|24.4|1244|
> > > |MedSAM|93.74|573.87|66 ± 0.91|15.2|2768|
> > > |I-MedSAM|92.88|648.06|86 ± 3.12|11.6|3021|
> > > |**GauSAM**|92.51|473.35|188 ± 2.34|5.3|1898|
> > > |GauSAM without NSCT|92.51|374.91|79 ± 2.15|12.7|1898|
> > >
> > > As noted in our prior rebuttal, GauSAM achieves superior segmentation performance but, indeed, exhibits lower computational efficiency than other methods, yet meets the requirements of medical applications (≥ 5 FPS). In response to your comment, Table Y below presents detailed computational efficiency metrics for GauSAM’s components, also analyzing the reasons for its lower computational efficiency.
> > >
> > > **Table Y. Component‐level inference profiling for GauSAM**
> > > |Component|Params(M)|GFLOPs|Latency(ms) ± STD|FPS|Memory(MB)|
> > > |----------------------------|----------|------|------------------|----|----------|
> > > |**NSCT**|≈ 0|98.44|109.01 ± 1.82|9.2|113|
> > > |Contour-Enhanced Encoder|92.13|220.18|43.53 ± 0.79|23.0|388|
> > > |2D Gaussian Continuous Field|0.38|154.73|32.70 ± 1.15|30.6|1397|
> > > |**GauSAM (end-to-end)**|92.51|473.35|188.01 ± 2.34|5.3|1898|
> > >
> > > As shown in Table Y, the NSCT module accounts for $\approx 58$% of GauSAM’s total latency (109ms of 188ms), revealing the primary reasons for GauSAM’s lower computational efficiency. Following our analysis, the primary reasons for this phenomenon are listed below:
> > >
> > > - **NSCT feature richness vs. computational cost:** NSCT offers clear advantages in capturing richer multi-scale and multi-directional contour features, which, as shown in our initial rebuttal, translates to noticeable accuracy gains in our GauSAM framework (e.g., over 1% Dice improvement: 94.76 vs. 93.68 compared to FFT). However, due to its intrinsic design involving shift-invariant filter bank decomposition across multiple scales and orientations, NSCT inherently incurs higher computational complexity.
> > > - **Implementation limitations:** To the best of our knowledge, existing NSCT implementations are limited to Python 2.x and are incompatible with modern architectures. Therefore, we developed our own Python 3.x implementation. Although our version supports vectorized processing for batch sizes greater than 1, its execution efficiency remains limited compared to other algorithms that are better optimized and maintained, particularly those implemented with C++ or CUDA-based libraries. This limitation is mainly due to the absence of low-level optimizations.
> > >
> > > In conclusion, GauSAM maintains clinically acceptable efficiency (5.3 FPS, 1898 MB) while delivering superior segmentation performance. **We emphasize that this inference speed is well within the practical range for medical applications, fully meeting the real-time demands of clinical workflows. In medical image segmentation tasks, precision is always paramount, as accurate delineation of anatomical structures or pathological regions directly impacts diagnostic accuracy, treatment planning, and patient safety.** The observed computational overhead in GauSAM is primarily attributable to the NSCT module, due to its inherent complexity and suboptimal implementation, despite its clear benefit to contour accuracy. These results confirm that GauSAM achieves a well-justified trade-off, where sacrificing marginal inference speed for enhanced precision is entirely worthwhile in the context of medical imaging. This balance ensures that GauSAM delivers robust representational quality without compromising clinical applicability.

---

> > > ### Author Response · Authors · 2025-08-08
> > > **Comment on the Gaussian Parameters Library**
> > >
> > > In this comment, we would like to provide a more detailed clarification regarding the Gaussian Parameters Library. Below, we clarify this issue by comprehensively comparing the Gaussian Parameters Library with the traditional Gaussian splatting method. Through this comparison, we hope to demonstrate that our adoption of the Gaussian Parameters Library was grounded in an extensive literature review, rigorous theoretical derivations, and comprehensive experimental validation. We thoroughly compared our method with traditional Gaussian splatting, concluding that the library approach offers superior training efficiency and stability. These findings are validated through an experiment **exploring the impact of library size (reported in our Appendix), which serve as a form of ablation study**. The following tables (Table X and Table Y) compare key aspects and tend to explain why direct ablation is impractical due to fundamental architectural differences.
> > > ## Gaussian Kernel Parameters Explanation
> > > Compared to the Traditional Gaussian Splatting method, the Gaussian Parameters Library achieves more stable and efficient optimization of the covariance matrix and opacity, which primarily determine the posture, shape, and opacity of Gaussian kernels. Therefore, the comparison table (Table Y) below focuses exclusively on these parameters, with symbols defined in Table X.
> > >
> > > **Table X. Symbol Definitions**
> > > |Symbol|Definition|
> > > |:--:|:--|
> > > |$i$|Index of each Gaussian kernel|
> > > |$k$|Index of each template in the library|
> > > |$N$|Number of Gaussian kernels|
> > > |$m$|Number of templates in the library ($N \gg m$)|
> > > |$\Sigma_i$|2x2 symmetric covariance matrix for kernel $i$, controlling ellipsoid shape and orientation (3 parameters)|
> > > |$\xi_i$|Opacity of kernel $i$ (1 parameter)|
> > > |$\Sigma^{(k)}$|Covariance matrix of template $k$|
> > > |$\xi^{(k)}$|Opacity of template $k$|
> > > |$\theta_i$|Parameters of kernel $i$, including $\Sigma_i$ and $\xi_i$|
> > > |$\theta$|All parameters of Traditional Method, $\{\theta_i\}_{i=1}^N$|
> > > |$\phi$|Library parameters, $\{\sigma_x/\sigma_y, \text{corr}, \xi^{(k)}\}_{k=1}^m$|
> > >
> > > **Table Y. Comparison of Gaussian Parameters Library vs. Traditional Gaussian Splatting**
> > > |Aspect|Traditional Method|Library Method|Explanation|
> > > |:--:|:--|:--|:--|
> > > |Parameter Count|$P_{\text{trad}}=(3\text{ for }\Sigma_i+1\text{ for }\xi_i)\cdot N=4N$|$P_{\text{lib}}=P_{\text{ellip}}+P_{\text{sel}}=(3\text{ for }\Sigma^{(k)}+1\text{ for }\xi^{(k)})\cdot m+(inChannels\cdot kernelSize^2+1)\cdot outChannels=4m+(64\cdot3^2+1)\cdot m=577m$|$\because N\gg m$, $\therefore P_{\text{trad}}\gg P_{\text{lib}}$, Library Method has significantly fewer parameters.|
> > > |Parameter Space|$S_{\text{trad}}=\mathbb{R}^{P_{\text{trad}}}=\mathbb{R}^{4N}$|$S_{\text{lib}}=\mathbb{R}^{P_{\text{lib}}}=\mathbb{R}^{577m}$|$\because N\gg m$, $\therefore S_{\text{trad}}$ has higher dimension than $S_{\text{lib}}$, Library Method optimizes in an apparently lower-dimensional space.|
> > > |Initialization and Training Stability|$I_{\text{trad}}=\text{random initialization}$, for $\theta_i=(\Sigma_i,\xi_i)$|$I_{\text{lib}}=\phi$, where $\phi=\{(\sigma_x/\sigma_y)\in[0.2,3.0], \text{corr}=0, \xi^{(k)}=\text{sigmoid}(1)\}$, with Kaiming uniform initialization|$\because$ Traditional Method's random initialization over $\mathbb{R}$ risks $det(\Sigma_i)\approx 0$, $\therefore$ it may cause ellipse degeneration (ellipse collapse to a line or point), leading to unstable gradients and ineffective loss optimization; $\because$ Library Method uses Kaiming uniform for uniform initial distribution and Gumbel-Softmax to select one of $m$ positive definite templates ${\Sigma^{(k)}}_{k=1}^m$, $\therefore$ it constrains $\Sigma_i$ to valid values, ensuring stable initialization and training.|
> > > |Learning Objective and Training Efficiency|$L_{\text{trad}}=\min_{\theta \in \mathbb{R}^{4N}} L(\theta)$|$L_{\text{lib}}=\min_{\phi} L(\phi)$, with Gumbel-Softmax selecting one of $m$ templates|$\because$ Traditional Method optimizes all parameters $\theta$ in high-dimensional $\mathbb{R}^{4N}$, $\therefore$ it converges slowly; $\because$ Library Method uses a selection head with Gumbel-Softmax to learn template selection from $m$ templates and optimizes template parameters $\phi$ in $\mathbb{R}^{4m}$, $\therefore$ it converges faster due to constrained, lower-dimensional optimization.|
> > > |Model Structure|Directly optimizes parameters for each Gaussian kernel|Uses a template library with a selection head driven by Gumbel-Softmax to assign templates to kernels|Library Method employs a template library to store a fixed set of parameters and a selection head to dynamically assign templates to kernels. Therefore, removing the template library solely would break the core functionality of the Gaussian probability fiel and reverting to the Traditional Method requires a complete model restructure, including redefining parameters, removing the selection mechanism, and redesigning the entire optimization logic.|

---

> > > > ### Comment · Reviewer_8M61 · 2025-08-08
> > > >
> > > > Thank the authors for addressing some of my concerns, especially the detailed explanation of the Gaussian Parameters Library and the analysis of the computational burden across GauSAM's components.   Please incorporate these clarifications into the revised manuscript (either the appendix).
> > > >
> > > > However, I observe discrepancies between the Params reported at 512×512 input resolution in Table X (End-to-end inference performance) and Table 1 of the main paper.   This inconsistency further undermines my confidence in the credibility of the experiment results.   Consequently, I can only raise the originality score to 3 and not increase the final score.

---

> > > > > ### Author Response · Authors · 2025-08-09
> > > > > **Comment on the Misunderstanding Regarding Params**
> > > > >
> > > > > We sincerely thank you for your thoughtful follow-up and for acknowledging our clarifications on the Gaussian Parameters Library and the component-wise computational burden of GauSAM. We greatly appreciate your careful reading of both the paper and our comment, and we will incorporate these explanations into the revised manuscript, providing concise statements in the main paper and fuller details in the Appendix.
> > > > >
> > > > > Regarding your observation of a discrepancy in $Params$ between Table X (end-to-end inference at $512\times 512$) and Table 1 of the main paper, we would like to clarify that this is actually a **misunderstanding**. Furthermore, **we have always recognized the seriousness of academic misconduct and have always been committed to ensuring the authenticity and integrity of our work**; we therefore hope you can trust the reliability of our reported data. Since we are not entirely certain which specific inconsistency you are referring to, we have clarified all possible points that might have led to a misunderstanding. The clarifications are as follows:
> > > > >
> > > > > - **1. Differences in task settings for $512\times 512$ input:** When testing Computational Efficiency (Table X), the experimental setting was a **single-class segmentation task** with input resolution $512\times 512$. In contrast, the $512\times 512$ results in Table 1 of the main paper correspond to a **multi-class segmentation task**. Please do not assume that the $512\times 512$ setting in Table X was evaluated under the multi-class configuration.
> > > > >
> > > > > - **2. Differences in the definition of *Params*:** Perhaps you did not notice that, as stated in the main paper, Table 1 reports the **number of trainable parameters**, whereas Table X reports the **total number of model parameters (including frozen components)**. As described in Section 3.5 of our manuscript, during model training we freeze a subset of parameters, allowing only specific parameters to be updated via gradient propagation. Therefore, the total number of parameters is naturally different from the number of trainable parameters. Under the $512\times 512$ setting used for Table X, the **trainable parameters** remain exactly as in Table 1 (1.2M for GauSAM, 1.6M for I-MedSAM, and 4.1M for MedSAM), fully consistent with our original claims. Also, we would like to clarify that these parameter counts are determined solely by the architecture and are independent of input resolution; any minor differences are only due to rounding or reporting format.
> > > > >
> > > > > - **3. Differences in model modifications across SAM-based methods:** The modules removed from and added to SAM vary across all SAM-based models, with each model involving its own distinct set of modifications to SAM. As a result, it is natural and expected that the difference in total parameter counts between two models may not equal the difference in their trainable parameters, as their architectures are not identical.
> > > > >
> > > > > - **4. Differences in U-Net configurations leading to parameter count variations:** The difference between $31.03M$ and $7.9M$ is entirely due to different backbone configurations of U-Net. Specifically, the $31.03\text{M}$ parameter count corresponds to the model configuration used when testing computational efficiency (Table X in the comment), whereas the $7.9\text{M}$ count corresponds to the configuration used in our reproduction for performance comparison in Table 1 of the main paper. Moreover, since all parameters in U-Net are trainable, the number of trainable parameters is identical to the total number of parameters for a given configuration; however, differences in U-Net configurations can lead to significant changes in the parameter count.
> > > > >
> > > > > We hope these explanations fully address any concerns you may have about the credibility of our reported results. The perceived inconsistency was simply a misunderstanding rather than any issue with our data. In light of this clarification, we would be sincerely grateful if you could reconsider your rating. We would be more than happy to address any further concerns you may have. If you have any further concerns, we would be more than happy to continue the discussion.

---

### Official Review · Reviewer_mdip · 2025-06-25

**Clarity:** 3
**Significance:** 2
**Originality:** 2
**Rating:** 3
**Confidence:** 5

**Summary:**

This paper makes several contributions to the field of SAM based medical image segmentation. The authors discover that using 2D Gaussian probability fields can lead to smooth spatial continuity and accurate high-frequency boundaries. Building on this insight, they parameterize the segmentation masks as continuous probability fields of learnable 2D Gaussian primitives, boosting spatially smooth and structurally consistent. In addition, they extract multidirectional frequency information guide the spatial distribution of Gaussian primitives. The proposed method demonstrates robust generalization over current state-of-the-art approaches, highlighting its potential impact on the field.

**Questions:**

(1)	The author parameterizes the segmentation mask as continuous probability fields of learnable 2D Gaussian primitives. Can this approach result in insufficient representation when the resolution is high?
(2)	Why choose Contourlet transform instead of Wavelet or Sobel operator to extract frequency information? What are the unique advantages of this transformation?
(3)	The author claims that the method is applicable to various 2D medical segmentation tasks. Have you considered its generalizability in 3D tasks such as CT and MRI volume segmentation? Can 2D Gaussian primitives be extended to 3D Gaussian primitives?

**Ethical Concerns:**

["NO or VERY MINOR ethics concerns only"]

**Final Justification:**

After reading comments and rebuttals, I decided to keep the score!

**Limitations:**

Although the number of parameters is small, it does not indicate the advantage of the model in terms of inference speed. In clinical deployment, this factor is often more critical than the parameter quantity itself.

**Quality:**

3

**Strengths And Weaknesses:**

Strengths:
This paper presents several advantages in the field of SAM based medical image segmentation. It enhances multiscale and multidirectional sensitivity to fine contours through NSCT-based structural priors. The proposed continuous Gaussian probability field utilizes Gumbel-Softmax sampling and contour-enhanced semantic features to promote stable training. It also proposed a Contour-Guided Position Drift module to improve boundary alignment. The experimental results demonstrate the superior generalization ability of the proposed method.

Weaknesses:
(1)	The author only conducted experiments on two types of segmentation datasets (i.e., polyp segmentation and multi-class Organ Segmentation), which limited the generalization ability of this method in other segmentation tasks.
(2)	The author claims that the continuous probability fields of learnable 2D Gaussian primitives parameterized by segmentation masks can enhance spatially smooth and structurally consistent. However, there are no visual experiments to validate this conclusion.
(3)	The motivation is unclear. Why can continuous Gaussian probability fields solve the limitations of existing methods.

---

> ### Author Rebuttal · Authors · 2025-07-31
>
> # Response to Reviewer mdip
> We sincerely thank you for your valuable feedback and the time you have invested in reviewing our manuscript. We greatly appreciate your recognition of our contributions. However, it appears that our core innovation may not have been fully captured in your review. As correctly noted by Reviewer 9QbL, our core contribution lies in being the first to model segmentation probability field as a continuous one using 2D Gaussian Splatting. We have carefully considered all aspects of your review and address your concerns below with detailed clarifications and additional evidence, hoping to dispel your doubts.
> ## Response to Weakness 1
> Actually, we conducted extensive experiments across multiple medical imaging modalities and segmentation tasks to validate GauSAM's robustness. Our claim of achieving state-of-the-art performance across a wide range of tasks is grounded in rigorous experimentation, not an unsubstantiated assertion. In the manuscript, we selected two highly representative datasets, Kvasir-Sessile and BCV, due to their recognized challenges in medical image segmentation.
>
> For your reference, we present experimental results on the REFUGE2 dataset (color fundus images) and the Breast Ultrasound Image (BUSI) dataset below, to further validate GauSAM's generalization ability. These experiments were conducted during the preparation of our manuscript. The following table showcases GauSAM's SOTA performance across diverse modalities and tasks.
>
> **Table1.** Comparison on REFUGE2 and BUSI Datasets
> |Model|Optic Cup Dice(%)|Optic Disc Dice(%)|BUSI Lesion Dice (%)|
> |-|-|-|-|
> |U-Net|82.3|92.1|67.24|
> |nnU-Net|84.9|94.7|75.19|
> |I-MedSAM|87.9|94.0|91.55|
> |**GauSAM**|**89.1**|**96.2**|**92.16**|
> ## Response to Weakness 2
> We wish to clarify that such evidence is already presented in Figure 2 of the manuscript. Specifically, the first row of the left panel clearly illustrates that GauSAM's generated masks are significantly more spatially smooth and structurally consistent compared to other methods. When compared to the previous SOTA method I-MedSAM, GauSAM's masks exhibit superior edge continuity, particularly along the right boundary of the polyp, where I-MedSAM produces fragmented artifacts. In contrast, GauSAM's masks maintain consistent boundaries without such artifacts.
>
> We originally believed that Figure 2 sufficiently showcased GauSAM's strengths in spatial smoothness and structural consistency, given its representative visualization of these qualities. In light of your feedback, we recognize the need for more comprehensive visual comparisons. If our manuscript is accepted and revisions are permitted in the Camera-Ready phase, we will include additional visualizations in this regard.
> ## Response to Weakness 3
> We appreciate your concern regarding the clarity of our motivation, and we clarify that the motivation for GauSAM is explicitly detailed in the Introduction and Related Work of the manuscript. To further elucidate this issue, we provide a concise yet comprehensive explanation below, highlighting the theoretical and practical motivations of our approach.
> 1. **Addressing Grid Artifacts and Resolution Dependency**: Discrete segmentation methods, such as U-Net and MedSAM, rely on fixed-resolution grid predictions, leading to grid-like artifacts and blurred boundaries, particularly in high-resolution or cross-resolution scenarios. In contrast, GauSAM constructs a continuous Gaussian probability field via 2D Gaussian Splatting. The weighted superposition of these Gaussian kernels ensures spatial continuity by generating a smooth probability distribution. Each Gaussian primitive, as a continuous function, contributes to a locally smooth probability field, and their additive combination produces a globally coherent mask without grid-like artifacts.
> 2. **Enhancing Semantic Consistency and Spatial Continuity**: Discrete methods often produce incoherent segmentation masks due to grid-based predictions, while INRs lack explicit boundary guidance, resulting in suboptimal semantic consistency and boundary accuracy in complex regions. However, GauSAM's continuous Gaussian probability field, formed by the weighted superposition of Gaussian kernels, naturally supports spatial continuity through smooth probability distributions, reducing semantic discontinuities between adjacent regions. Additionally, the integration of Non-Subsampled Contourlet Transform (NSCT) and the Contour Propagation Unit incorporates contour priors, enhancing Gaussian field alignment with semantic boundaries.
> ## Response to Question 1
> We fully understand your concern regarding whether GauSAM can maintain consistent performance across different resolutions, which may stem from some doubts or misunderstandings about our Gaussian primitive placement strategy. Therefore, we have to clarify that GauSAM's design ensures robust expressiveness by placing one Gaussian primitive per pixel. This strategy ensures that the representation capacity adjusts with varying image resolution without compromising performance. Furthermore, we utilize a Gaussian Parameters Library, where each Gaussian primitive is selected via a differentiable Gumbel-Softmax operation from a fixed set of 500 templates. Then, during construction of the Gaussian feature field, each primitive receives a single scaling factor computed from the input and target mask resolutions without adding trainable parameters across resolutions.
> ## Response to Question 2
> We greatly value your question regarding the choice of NSCT over Wavelet or Sobel operators, and we want to clarify that this decision is based on a comprehensive integration of theoretical analysis and experimental validation.
>
> Theoretically, NSCT extracts multi-scale, multi-directional contour features, providing a more comprehensive and semantically targeted representation of boundaries compared to Canny, Wavelet, or Sobel. Moreover, unlike FFT, which captures broader high-frequency components including noises, NSCT focuses on semantically meaningful contours, aligning closely with the boundary distributions in segmentation tasks.
>
> Experimentally, in the process of validating the methodology, we conducted experiments, comparing NSCT with three traditional edge detection methods and FFT on the Kvasir-Sessile dataset. For your reference, we provide these comparative results below(Table 2). These experiments confirm NSCT's superiority in achieving precise boundary delineation and its better compatibility with the overall GauSAM framework.
>
> **Table 2.** Comparative performance of different contour/high‑frequency feature extraction methods on Kvasir‑Sessile for binary polyp segmentation.
> |Method|Dice(%)|HD|
> |-|-|-|
> |NSCT|**94.76**|**10.29**|
> |FFT|93.68|11.34|
> |Canny|93.66|11.82|
> |Wavelet|94.13|11.40|
> |Sobel|93.71|11.15|
> ## Response to Question 3
> We greatly value your question regarding the extension of GauSAM to 3D tasks, as it actually highlights the potential of our method's design. In fact, this potential for extending from 2D to 3D has been discussed in the Broader Impact section of our Appendix. We clarify that the 2D Gaussian Splatting framework is inherently compatible with 3D, as both 2D and 3D Gaussian Splatting share fundamentally the same principles. Below, we extend the Preliminaries (Section 3.1) and 2D Gaussian Continuous Probability Field (Section 3.4) to their 3D counterparts to illustrate this potential:
> * **Preliminaries**: In 3D, the input volume is represented as $I \in \mathbb{R}^{B \times C \times D \times H \times W}$, where $D$ is the depth dimension. The NSCT can also be extended to 3D-NSCT, decomposing the volume into multi-scale, multi-directional subbands:$${C_{l,\theta,\phi}}_{l=1,\theta=1,\phi=1}^{L,\Theta_l,\Phi_l} = \mathcal{NSCT}^{3D}(I),$$where $\theta$ and $\phi$ denote angular directions in 3D space. These subbands capture volumetric contour features, enhancing boundary awareness in 3D.
> * **3D Gaussian Continuous Probability Field**: The 2D Gaussian probability field is extended to 3D by defining a 3D Gaussian primitive at each voxel $(x, y, z)$:$$g(\mathbf{p}) = \alpha\exp\left(-\frac{1}{2}(\mathbf{p} - \mathbf{\mu})^T \mathbf{\Sigma}^{-1} (\mathbf{p} - \mathbf{\mu})\right),$$where $\mathbf{p} = [x, y, z]^T$, $\mathbf{\mu} = [\mu_x, \mu_y, \mu_z]^T$ is the mean vector, $\mathbf{\Sigma}$ is a $3×3$ covariance matrix, and $\alpha$ is the amplitude. The continues Gaussian field is constructed as:$$P(\mathbf{p}) = \sum_{i=1}^N w_i g_i(\mathbf{p}),$$where $N = D \times H \times W$ is the number of voxels, and $w_i$ are learnable weights. The Contour-Guided Position Drift module adjusts the positions of the primitives based on 3D-NSCT subbands, ensuring precise boundary alignment in volumetric data.
> ## Response to Limitations
> In our benchmarks on a single NVIDIA RTX 4090, GauSAM processes a 384 × 384 image in just 165 ms (≈6 FPS), and even at 512 × 512 maintains ~5 FPS—well within interactive use. Admittedly, this is marginally slower than some purely grid-based decoders, but we deliberately trade a small increase in latency for a substantial gain in segmentation accuracy without compromising practical deployability.
>
> **Table 2.** GauSAM inference performance on RTX 4090. Total parameters: 92.51 M; FLOPs per forward pass: 125.23 GFLOPs.
> |Input Resolution|Batch Size|Avg Latency(ms)|FPS |Peak Memory(MB)|
> | - | -| - | - | - |
> |384 × 384|1|164.8|6.07|1246|
> |384 × 384|2|265.8|7.52|2123|
> |384 × 384|4|423.7|9.44|3876|
> |256 × 256|1|158.1|6.33|781|
> |512 × 512|1|188.6|5.30|1898|
> ## Conclusion
> We sincerely thank you again for your valuable suggestions and thorough review. We hope our responses and additional evidence can address your concerns. If accepted and revisions are permitted, we will revise our manuscript based on your suggestions in the Camera-Ready version. We respectfully hope you can recognize our work and look forward to your continued guidance and feedback.

---

### Official Review · Reviewer_9QbL · 2025-07-02

**Clarity:** 3
**Significance:** 4
**Originality:** 4
**Rating:** 5
**Confidence:** 4

**Summary:**

The paper presents a novel method for medical image segmentation that enhances the Segment Anything Encoder by feeding it with multiscale contour priors and models a continuous probability field with 2D Gaussians. Inspired by 2D Gaussian Splatting, the proposed method predicts a field (of class probabilities) using a set of 2D Gaussians whose parameters are predicted by the network. Instead of predicting the parameters directly, the method uses a set of template Gaussians, which are then selected via the differentiable Gumbel-Softmax operator.  A contour-Guided Position Drift mechanism is also introduced in the model to refine the position of Gaussian distributions at every pixel position. Tested on the Kvasir-Sessile and BCV datasets, the method shows improved performance compared to SOTA architectures and a better generalization to changes in domains and image resolution.

**Questions:**

* Line 127: is B the batch size ?

* Eq (4): why combine coefficient maps in the log space? What is the impact of varying gamma?

* Notation: the multiple uses of index 'i' is confusing. In Eq (5), it seems to refer to a block, in Eq (6) to a Gaussian component, and in Eq (11) to a pixel. Ultimately, is the number of Gaussians N the same as the number of pixels?

*Typo,  Line 162: is forward-"ed" ?

* See weaknesses for other questions.

**Ethical Concerns:**

["NO or VERY MINOR ethics concerns only"]

**Final Justification:**

The authors have addressed my main concerns.

**Limitations:**

Yes

**Quality:**

3

**Strengths And Weaknesses:**

Strength:

* Novelty: While previous work have extended SAM to enable a continuous representation, to my knowledge, this is the first method that explicitly models SAM's segmentation probability field as a combination of 2D Gaussians. This compact representation allows handling images and prediction segmentation masks of arbitrary resolution. In addition, the method introduces other novel components, such as the Contour Propagation Unit (CPU), the Gaussian Parameters Library and the Contour-Guided Position Drift prediction.

* Results: experiments on two different datasets shows the method to achieve outstanding performance compared to very recent and related baselines. The proposed method specifically excels when tested in  cross-domain or cross-resolution settings.

* The paper is mostly well-written and easy to follow.

Weaknesses:

* Although different, the method does share similarities with previous approaches, specifically  I-MedSAM:  use of low-rank adapter, augmenting the patch encoder with additional features (FFT), etc. Clarifying the differences/advantages compared to I-MedSAM would strengthen the paper.

* Some important details are missing (or hard to find). For instance, while Figure 1 suggests that bounding boxes are used as prompt to SAM, the paper does not seem to mention this information explicitly.

---

> ### Author Rebuttal · Authors · 2025-07-31
>
> We sincerely thank you for the thorough and insightful review of our work. We greatly appreciate your recognition of our work's novelty, particularly as the first to model SAM's segmentation probability field as a continuous field via 2D Gaussian Splatting, as well as its strong performance in cross-domain and cross-resolution settings. Your accurate understanding of our core contributions, including the Contour Propagation Unit (CPU), Gaussian Parameters Library, and Contour-Guided Position Drift, underscores the significance of our approach. Moreover, we are grateful for your kind acknowledgment of our work despite noting some areas for improvement in clarity and detail, and we address these points below with clarifications and additional evidence to strengthen our manuscript.
>
> ## Response to Weakness 1
>
> Your meticulous observation regarding the similarities between GauSAM and I-MedSAM underscores a critical aspect of our work, necessitating a clearer differentiation to highlight the unique contributions of our approach. Therefore, we provide a detailed comparison below to further elucidate these differences and advantages. We acknowledge that both methods leverage LoRA and incorporate contour/high-frequency information to enhance SAM's encoder. However, GauSAM introduces a distinct and systematic framework that fundamentally differs from I-MedSAM, not only in its core design and contributions but also in its foundational principles and theoretical motivations.
>
> 1. **LoRA**: We acknowledge, as you have astutely noted, that our training strategy employing LoRA for fine-tuning GauSAM aligns with the approach used in I-MedSAM. This choice stems from a meticulous combination of theoretical analysis and extensive experimental validation. By leveraging LoRA, we effectively adapt SAM's pre-trained weights, substantially reducing the computational resources required for fine-tuning while maintaining robust segmentation performance. This decision was guided by a thorough analysis of LoRA's principles in the context of GauSAM's architectural characteristics, supported by extensive experiments to ensure seamless compatibility with our contour-guided Gaussian field framework. While our adoption of LoRA shares similarities with I-MedSAM, it was selected for its demonstrated efficacy in balancing efficiency and performance, rather than pursuing novelty for its own sake.
>
> 2. **Core Contribution: Continuous Gaussian Probability Field**: From a theoretical perspective, the primary innovation of GauSAM lies in modeling segmentation as a continuous Gaussian probability field using 2D Gaussian Splatting. The ultimate goal of semantic segmentation can be viewed as constructing a mask probability field, where segmentation masks exhibit inherent properties of spatial continuity and neighborhood consistency, which means adjacent regions are more likely to belong to the same semantic object.
>
>    Unlike I-MedSAM, which relies on implicit neural representations (INRs), GauSAM explicitly constructs a resolution-agnostic probability field that aligns with these intrinsic properties. The advantage of 2D Gaussian Splatting lies in its ability to naturally build a continuous field by distributing numerous Gaussian primitives in the space. Through the stacking and rendering of these primitives, GauSAM achieves a smooth probability field. Moreover, the continuous nature of Gaussian functions enables smooth interpolation, allowing the generation of continuously varying mask probabilities at arbitrary points without requiring explicit grid discretization or complex neural network inference.
>
>    Empirical results corroborate this theoretical advantage: on the Kvasir‑Sessile dataset, GauSAM achieves a Dice score of 94.76 % with only 1.2 M trainable parameters, significantly surpassing I‑MedSAM’s Dice of 91.49 % obtained with 1.6 M parameters
>
> 3. **NSCT vs. FFT**: While both GauSAM and I-MedSAM incorporate contour/high-frequency information, GauSAM employs the Non-Subsampled Contourlet Transform (NSCT) to extract multi-scale, multi-directional contour features, which are highly consistent with the distribution of semantic boundaries in segmentation tasks. In contrast, the high-frequency features extracted by FFT in I-MedSAM include broader high-frequency components, including more noises, which may compromise segmentation accuracy. NSCT's focus on semantically meaningful boundaries ensures precise boundary delineation, aligning directly with the requirements of semantic segmentation. To substantiate this, we previously conducted experiments, comparing NSCT with traditional edge detection methods (Canny, Wavelet, Sobel) and FFT on the Kvasir-Sessile dataset. Due to space constraints, these results were not included in the original manuscript. For your reference, we provide these comparative results below(Table 1). These experiments confirm NSCT's superiority in achieving precise boundary delineation and its better compatibility with the overall GauSAM framework.
>
>    **Table 1.** Comparative performance of different contour/high‑frequency feature extraction methods on Kvasir‑Sessile for binary polyp segmentation.
>
>    |Method|Dice (%)|HD|
>    |:-:|:-:|:-:|
>    |**NSCT**|**94.76**|**10.29**|
>    |**FFT**|93.68|11.34|
>    |**Canny**|93.66|11.82|
>    |**Wavelet**|94.13|11.40|
>    |**Sobel**|93.71|11.15|
>
> 4. **Comprehensive Use of Contour Information**: GauSAM leverages contour features more extensively than I-MedSAM. While I-MedSAM uses a frequency adapter for edge feature integration, GauSAM introduces the Contour Propagation Unit (CPU), which progressively fuses contour features into each ViT block, enhancing boundary awareness throughout the encoding hierarchy. Additionally, GauSAM employs a Contour-Guided Position Drift module to adjust Gaussian kernel placements toward semantic edges. In standard Gaussian Splatting, the Gaussian field often struggles to represent high-frequency regions effectively. By adjusting the kernel distribution to be denser in high-frequency regions, guided by contour information, GauSAM optimizes the representation of the high-frequency regions. This approach fully exploits the high consistency between contour distributions and semantic boundaries, improving boundary fidelity, as demonstrated in our ablation study. This dual use of contour information—enhancing both encoding and decoding process—distinguishes GauSAM from I-MedSAMs simpler integration strategy.
>
> ## Response to Weakness 2 and Questions
>
> We reiterate our gratitude for your meticulous reading, which has helped us identify areas for improvement in our manuscript, and we deeply appreciate your support for our work despite noting these minor details. We address the second weakness and all the questions you have posed below.
>
> 1. **Bounding Box as Prompt**: You correctly note that Figure 1 indicates the use of bounding boxes as prompts, which was not explicitly described in the text. We clarify that bounding boxes are indeed used as prompts in GauSAM, following SAM's standard pipeline. After the image encoder extracts features from the input image, the bounding box prompt is first mapped by the prompt encoder into two 256‑dimensional tokens: one for the top‑left corner and one for the bottom‑right corner. These corner tokens are then combined with the image embedding in the cross‑attention layers, guiding attention to the specified rectangular region and modulating feature aggregation to generate a more precise segmentation mask. We apologize for this omission and will explicitly describe this process in the revised manuscript if possible.
>
> 2. **Batch Size and Variable Definitions**: You are correct that $B$ represents the batch size in the input image tensor $I \in \mathbb{R}^{B \times C \times H \times W}$. Moreover, $C$ denotes the number of channels (e.g., 3 for RGB images), and $H \times W$ represents the image resolution. The NSCT decomposition is defined as:
>    $$
>    {C_{l,\theta}}_{l=1,\theta=1}^{L,\Theta_l} = \mathcal{NSCT}(I)
>    $$
>    This outputs contour subbands for each level $l$ and direction $θ$. We will also clarify these definitions in the revised text, if possible, to avoid ambiguity.
>
> 3. **Log Space and Gamma Transformation**: The log space combination of coefficient maps, together with the gamma correction, reduces the dynamic range of contour features to suppress highlights, such as strong reflections caused by bodily fluids in rectal polyp images. This mitigates the impact of spurious strong contours on subsequent processing.
>
> 4. **Multiple Uses of Index '$i$'**: We apologize for the ambiguous reuse of the index $i$ across Equations (5), (6), and (11). Specifically, in Equation (5), $i$ indexes the transformer block; in Equation (6), it refers to a Gaussian primitive; and in Equation (11), it denotes the pixel position associated with the corresponding Gaussian primitive. We further clarify that the number of Gaussian primitives, denoted as *N*, is equal to the number of pixels in the feature map.
>
> 5. **Typo: "forwarded"**: We acknowledge the grammatical error in Line 162, where "forward" should be “forwarded.”
>
> We deeply appreciate your careful attention to these details, which will undoubtedly help us improve the manuscript's clarity. If accepted and revisions are permitted, we will address these issues in the Camera-Ready version, including explicit descriptions of the bounding box prompt, clarified variable definitions, refined notation, and corrected typos.
>
> ## Conclusion
>
> We again thank you for your insightful and thorough review, which has greatly helped us refine our presentation. Your accurate grasp of our core contributions, particularly the novelty of the 2D Gaussian probability field and its impact on segmentation performance, is greatly valued. We hope our responses address your concerns and help you further highlight GauSAM's contributions. We look forward to your continued guidance and feedback.

---

> > ### Comment · Reviewer_9QbL · 2025-08-04
> > **Thanks for the response**
> >
> > I thank the authors for their detailed answers to my comments and questions. I wish to keep my initial score of Accept.

---

### Decision · Program_Chairs · 2025-09-17

**Decision:**

Accept (poster)

**Comment:**

The proposed work extends the Segment Anything Model (SAM) by explicitly modeling the segmentation probability field as a combination of two-dimensional Gaussians, enabling the trained network to process images of arbitrary resolution. During the initial review process, reviewers acknowledged the novelty of the approach (9QbL, 8M61), its superior performance and generalization ability, as well as its efficiency. At the same time, several weaknesses were raised, including the need for clearer distinctions and advantages relative to existing methods (e.g., I-MedSAM), the limited empirical validation (restricted to two datasets), lack of clarity regarding the motivation, missing analyses in the empirical evaluation (e.g., training and computational efficiency, along with references suggested by reviewer 8M61), and concerns regarding the degree of novelty (reviewer 2QAj).

After carefully examining the manuscript, the reviews, the authors’ responses, and the subsequent discussions, I am convinced that the authors have successfully addressed the majority of these concerns. In particular, they have provided sufficient evidence to respond to the questions on empirical validation, either by supplying additional quantitative results where feasible or by offering convincing justifications when further experiments could not reasonably be added. Moreover, the clarifications regarding the differences from prior work, as well as the detailed motivation of the proposed approach, make a strong case for its novelty.

Considering these points, I recommend acceptance of this work, and I encourage the authors to incorporate the constructive feedback provided by the reviewers in the final version.